# Inflammation status modulates the effect of host genetic variation on intestinal gene expression in inflammatory bowel disease

Shixian Hu[1,2,6], Werna T. Uniken Venema [1,2,6], Harm-Jan Westra[2], Arnau Vich Vila[1,2], Ruggero Barbieri [1,2], Michiel D. Voskuil[1], Tjasso Blokzijl[1], Bernadien H. Jansen[1], Yanni Li [1,2], Mark J. Daly [3,4], Ramnik J. Xavier [3,5], Gerard Dijkstra [1], Eleonora A. Festen[1,2,7] & Rinse K. Weersma[1,7 ✉]

More than 240 genetic risk loci have been associated with inflammatory bowel disease (IBD), but little is known about how they contribute to disease development in involved tissue. Here, we hypothesized that host genetic variation affects gene expression in an inflammation-dependent way, and investigated 299 snap-frozen intestinal biopsies from inflamed and non-inflamed mucosa from 171 IBD patients. RNA-sequencing was performed, and genotypes were determined using whole exome sequencing and genome wide genotyping. In total, 28,746 genes and 6,894,979 SNPs were included. Linear mixed models identified 8,881 independent intestinal *cis*-expression quantitative trait loci (*cis*-eQTLs) (FDR < 0.05) and interaction analysis revealed 190 inflammation-dependent intestinal *cis*-eQTLs (FDR < 0.05), including known IBD-risk genes and genes encoding immune-cell receptors and antibodies. The inflammation-dependent *cis*-eQTL SNPs (eSNPs) mainly interact with prevalence of immune cell types. Inflammation-dependent intestinal *cis*-eQTLs reveal genetic susceptibility under inflammatory conditions that can help identify the cell types involved in and the pathways underlying inflammation, knowledge that may guide future drug development and profile patients for precision medicine in IBD.

[1] Department of Gastroenterology and Hepatology, University of Groningen and University Medical Center Groningen, Groningen, The Netherlands. [2] Department of Genetics, University of Groningen and University Medical Center Groningen, Groningen, The Netherlands. [3] Broad Institute of Harvard and Massachusetts Institute of Technology, Cambridge, MA, USA. [4] Institute for Molecular Medicine Finland, HiLIFE, University of Helsinki, Helsinki, Finland. [5] Center for Microbiome Informatics and Therapeutic, Massachusetts Institute of Technology, Cambridge, MA, USA. [6] These authors contributed equally: Shixian Hu, Werna T. Uniken Venema. [7] These authors jointly supervised this work: Eleonora A. Festen, Rinse K. Weersma. ✉email: r.k.weersma@umcg.nl

nflammatory bowel disease (IBD), consisting of Crohn's disease (CD) and ulcerative colitis (UC), is an immune-mediated disorder characterized by chronic inflammation of the gastrointestinal tract. The etiology of IBD is not fully understood, and no cure is available, with current treatments only showing long-term effectiveness in a minority of patients[1]. Moreover, the prevalence of IBD is rising in westernized countries[2], highlighting the need to better understand the disease.

IBD is a genetically complex disease, and >240 IBD susceptibility loci have been identified to date[3]. Genomic variants within these loci are expected to play a role in the disease pathology, but clear pathomechanisms have been investigated for only a minority of risk genes, and causality has been proven for just a few[4]. Genetic variants can influence the transcription of close-by genes through *cis*-expression quantitative trait loci (*cis*-eQTL). To date, *cis*-eQTL studies have primarily been performed on general population cohorts, which have revealed regulatory modules within IBD risk loci that point to putative candidate genes within these loci[5–8]. However, while it is increasingly recognized that *cis*-eQTLs can be tissue-specific[9,10], the number of studies on intestinal tissues in IBD is low, and studies on *cis*-eQTLs in the context of intestinal inflammation in IBD are particularly scarce and limited in size.

Here, we hypothesized that the effect of host genetics on gene expression in IBD is dependent on inflammation status. To address this, we studied inflammation-dependent *cis*-eQTLs in IBD using RNA-sequencing (RNA-seq) data from 299 inflamed and non-inflamed snap-frozen intestinal biopsies derived from 171 IBD patients (Table 1) for whom we also have whole-exome sequencing (WES) data combined with genome-wide screening array (GSA) data to enlarge the reference. We then explored the functional impacts of these *cis*-eQTL effects by investigating regulatory effects and cell type involvement. Taken together, we pinpoint that intestinal mucosa *cis*-eQTLs can depend on the inflammatory status and be influenced by the cell type composition of the mucosa.

## Results

After QC and filtering ("Methods"), we analyzed genotype data of 165 individuals with IBD (UC = 68, CD = 97) and messenger RNA (mRNA)-sequencing data of 280 intestinal mucosal biopsy samples from these patients: 112 samples from inflamed tissue and 168 from non-inflamed tissue.

**Inflamed and non-inflamed intestinal areas show differential gene expression**. Comparing inflamed tissue to non-inflamed tissue revealed 1131 differentially expressed genes (linear regression, $t$ test, false discovery rate (FDR) < 0.001, $\beta > 1.5$; Supplementary Data 1a). Among the top genes in the inflamed biopsies are CXC chemokines such as *CXCL1* and *CXCL3*, which are chemoattractants for neutrophils, and *CXCL2*, which suppresses cell proliferation of hematopoietic progenitors. Overall, genes that are upregulated in inflamed intestinal tissue are enriched for cell communication and interaction pathways such as "interleukin-4 and -13 signaling" ($P$ value = 4.54e − 96) and "GPCR ligand binding" ($P$ value = 2.88e − 39) and for pathways involved in reorganization of extracellular matrix such as "extracellular matrix degradation" ($P$ value = 1.78e − 48). Genes downregulated in inflamed tissue are involved in (membrane) transport through pathways such as "SLC-mediated transmembrane transport" ($P$ value = 7.55e − 96; Supplementary Data 1b). We performed cell type deconvolution using xCell[11] and assessed 28 reference cell types thought to be present in the intestinal mucosa ("Methods"). This showed an enrichment of M1 macrophages, plasma cells, neutrophils, T-helper type 2 (Th2) cells

and T-regulatory cells (Tregs) in the inflamed tissue, whereas the non-inflamed tissue showed a relative enrichment of M2 macrophages and basophils (two-sided Wilcoxon's test, FDR < 0.05; Supplementary Fig. 1). When comparing gene expression between different disease groups (UC = 107, CD = 173 biopsies), we found 212 differentially expressed genes (linear regression, $t$ test, FDR < 0.001, $\beta > 1.5$) between CD and UC samples (Supplementary Data 2). Comparing transcriptional profiles in the colon vs. the ileum (colon = 191, ileum = 89 biopsies)

### Table 1 Cohort description.

| | Inflamed dataset | Non-inflamed dataset | P values |
|---|---|---|---|
| Inflammation, no. (%) | | | |
| Inflamed | 112 [100] | 0 | |
| Non-inflamed | 0 | 168 [100] | |
| Location, no. (%) | | | |
| Colon | 81 [72] | 110 [65] | $P_{\chi^2} = 0.28$ |
| Ileum | 31 [28] | 58 [35] | |
| Diagnosis, no. (%) | | | |
| Crohn's disease | 63 [56] | 110 [65] | $P_{\chi^2} = 0.29$ |
| IBD undefined | 12 [11] | 15 [9] | |
| Ulcerative colitis | 37 [33] | 43 [26] | |
| Sex, no. (%) | | | |
| Female | 68 [61] | 103 [61] | $P_{\chi^2} = 1$ |
| Age at biopsy, mean ± SD | 42.7 ± 15.1 | 42 ± 15.1 | $P_{\text{Wilcoxon}} = 0.61$ |
| Medication, yes (%) | | | |
| Mesalazines | 36 [32] | 49 [29] | $P_{\chi^2} = 0.69$ |
| Steroids | 26 [23] | 31 [18] | $P_{\chi^2} = 0.38$ |
| Thiopurines | 31 [28] | 46 [27] | $P_{\chi^2} = 0.95$ |
| Methotrexate | 1 [1] | 4 [2] | NA |
| Anti-TNF | 20 [18] | 37 [22] | $P_{\chi^2} = 0.51$ |
| Montreal classification, no. (%) | | | |
| Within CD | | | |
| Montreal A | | | |
| A1: <17 years | 9 [14] | 14 [13] | $P_{\chi^2} = 0.23$ |
| A2: 17–40 years | 40 [63] | 81 [74] | |
| A3: >40 years | 14 [22] | 15 [14] | |
| Montreal L | | | |
| L1 (+L4) | 9 (2) [14 (3)] | 23 (3) [21 (3)] | $P_{\chi^2} = 0.23$ |
| L2 (+L4) | 14 (2) [22 (3)] | 17 (2) [15 (2)] | |
| L3 (+L4) | 29 (7) [46 (11)] | 55 (10) [50 (9)] | |
| Montreal B | [1 missing] | [1 missing] | |
| B1 (p) | 17 (11) [27 (17)] | 37 (17) [34 (16)] | $P_{\chi^2} = 0.49$ |
| B2 (P) | 16 (9) [25 (14)] | 23 (16) [21 (15)] | |
| B3 (P) | 6 (3) [10 (5)] | 7 (9) [6 (8)] | |
| Within UC + IBDU | | | |
| Montreal E | [5 missing] | [4 missing] | |
| E1 | 2 [5] | 2 [4] | $P_{\chi^2} = 0.93$ |
| E2 | 15 [34] | 17 [31] | |
| E3 | 27 [61] | 35 [65] | |
| Montreal S | [11 missing] | [15 missing] | $P_{\chi^2} = 0.93$ |
| S1 | 0 | 1 [2] | |
| S2 | 6 [16] | 7 [16] | |
| S3 | 20 [53] | 23 [53] | |
| S4 | 12 [32] | 12 [28] | |

$P$ values of comparing categorical variables between groups are from two-sided $\chi^2$ test; $P$ values of comparing continuous variable between two groups are from two-sided Wilcoxon's test.

identified 2145 differentially expressed genes (linear regression, $t$ test, FDR < 0.001, $\beta$ > 1.5; Supplementary Data 3). Genes over-expressed in the ileum included *SLC28A1*, which is involved in transport of nutrients, and *MALRD1*, which has a function in bile acid regulation.

**Intestinal *cis*-eQTLs in IBD largely overlap with non-disease intestinal *cis*-eQTLs.** We first set out to investigate general *cis*-eQTL effects in intestinal tissue from patients with IBD that are independent of disease subtype, biopsy location, and inflammation. This identified 8881 unique genes (eGenes) with a *cis*-eQTL effect in human intestinal mucosal tissue (linear regression, $t$ test, FDR < 0.05; Supplementary Data 4a).

We first compared each significant gene—single-nucleotide polymorphism (SNP) pair (FDR < 0.05) from our dataset with all significant *cis*-eQTL results from the healthy intestinal datasets of the GTEx project[12] and the "CEDAR" cohort study[5]. GTEx provides three different RNA-seq-based sources of intestinal *cis*-eQTLs: sigmoid, transverse colon, and terminal ileum. Our eQTLs overlap with 97.48% (1085 out of 1113) of the GTEx sigmoid *cis*-eQTLs, with 99.27% (1778 out of 1791) of the GTEx transverse colon *cis*-eQTLs and 99.15% (937 out of 945) of the GTEx terminal ileum *cis*-eQTLs (Supplementary Fig. 2A–C). The replication rates in "CEDAR" cohort study, which provides three array-based intestinal *cis*-eQTL datasets[5], are 92.86% (65 out of 70 in the ileum), 92.25% (119 out of 129 in the transverse colon), and 94.64% (106 out of 112 in the rectum) (Supplementary Fig. 2D–F). We then compared the eQTLs reported here with those found in the pediatric IBD "RISK" cohort study[13] ($P$ < 0.05), a targeted eQTL study on known IBD GWAS variants. We found that 83.00% of *cis*-eQTLs have the same direction of effect (39 out of 47; Supplementary Data 4b). In addition, we compared our *cis*-eQTL pairs with the findings of the eQTLGen[14] meta-analysis, which was performed on blood and found a replication rate of 81.44% (Supplementary Fig. 2G), suggesting tissue-specific genetic regulatory effects to exist in our findings.

Interestingly, six eSNP–eGene pairs have different directions of effect as compared to the GTEx study (data were not present in the "CEDAR" study). After a heterogeneity test between these eQTL pairs and the three GTEx gut datasets, four eQTL pairs showed significance, which suggests that these intestinal *cis*-eQTLs indeed have a different direction of effect in our dataset in the context of IBD ($Q$ test $P$ < 0.05; Supplementary Data 4c and Supplementary Fig. 3). These four eGenes consist of: *PPP2R2D*, a gene involved in the cell cycle by controlling mitosis entry and exit; *RBL2*, a gene associated with type 2 diabetes; *LIMD1*, a gene involved in several cellular processes including cell–cell adhesion and cell development; and *ZNF593*, which modulates DNA binding. Neither the eGenes nor the eSNPs have previously been reported to be associated with IBD risk.

**Genomic variants within disease susceptibility loci affect intestinal gene expression.** To explore the functional impact of intestinal *cis*-eQTLs in gut diseases, we extracted GWAS summary statistics for six disease traits, (1) IBD, (2) CD, (3) UC, (4) colon cancer, (5) diverticulitis, and (6) coeliac disease. Using "coloc"[15], we performed colocalization analysis of the identified *cis*-eQTLs and disease GWAS loci to identify potential shared causal variants. At a posterior probability threshold of having one shared causal variant (PP4) of >0.5, we discovered 558 colocalizing variants (Supplementary Data 5). For example, our IBD-based dataset showed 172 eSNPs that colocalized with IBD. The eGene that most strongly colocalized with IBD is *HNF4A* (PP4 = 0.99), the expression of which is known to be decreased in the intestinal mucosa in patients with IBD and UC[16]. Functional

enrichment analysis showed that the eGenes that colocalized with IBD GWAS loci are enriched for the "olfactory signaling pathway" ($P$ value = 1.4e − 08) and "G alpha (s) signaling events" ($P$ value = 1.6e − 07). Both of these pathways are forms of G protein-coupled receptor signaling, which is a basic mechanism in the immune response in IBD[17]. For colon cancer, we found four colocalizing eSNPs, and for diverticulitis, we found one colocalizing eSNP. One hundred and two eSNPs colocalize with coeliac disease, which are enriched for the "ER–phagosome pathway" ($P$ value = 3.3e − 06) and "nucleotide excision repair" ($P$ value = 7.5e − 06). ER stress pathways are known to play a central role in IBD inflammation[18]. These results suggest that a large part of intestinal eSNPs are likely to be causal variants in IBD and coeliac disease.

**Intestinal *cis*-eQTLs are inflammation-dependent.** We then set out to identify genomic variants that differentially affect gene expression in the presence of intestinal inflammation in IBD. We, therefore, performed a genetics × inflammation-interaction analysis ("Methods"), which revealed 1854 inflammation-dependent *cis*-eQTLs (linear regression, $t$ test, $FDR_{interaction}$ < 0.05), involving 190 unique eGenes (Supplementary Data 6a, b). Subsequently, we determined whether these interactions could have been observed by chance. After ten permutations, the number of interaction eQTLs with an interaction $P$ value below the threshold associates with an FDR <0.05 in the non-permuted data, suggesting that the FDR estimates are well calibrated for the 1854 inflammation-dependent *cis*-eQTLs (Supplementary Fig. 4). Among these eGenes are *MIR214*, associated with progression of UC[19], *C6*, a complement protein encoding gene, and the gene encoding *FOLR3*, an antimicrobial and antitumor functioning protein[20].

The most significantly associated eSNP for each of the 190 eGenes was selected for further analysis. By assessing the significance of genotype and interaction items in the model, we found that 166 eGenes are mainly driven by interaction between eSNPs and inflammation status (linear regression, $t$ test, $FDR_{genotype}$ > 0.05, $FDR_{interaction}$ < 0.05). One inflammation-dependent eSNP is located close to IBD risk-associated genes: the eSNP rs12582553 (C/T, linear regression, $t$ test, $FDR_{interaction}$ = 0.046, $FDR_{genotype}$ = 1) is positioned upstream of its eGene *IL26* (Fig. 1a), which encodes an inflammatory mediator[21]. The gene *TREML4*, involved in Toll-like receptor signaling in macrophages[22], is upregulated by eSNP rs4337930 (T/C, linear regression, $t$ test, $FDR_{interaction}$ = 0.00046, $FDR_{genotype}$ = 1) under inflammatory circumstances, while its expression is independent of genotype in non-inflamed tissues. We also found that the eSNP rs3808491 (C/T, linear regression, $t$ test, $FDR_{interaction}$ = $7.23 \times 10^{-6}$, $FDR_{gentoype}$ = 1) forms an inflammation-dependent *cis*-eQTL with its eGene, *LY6D*, which is upregulated in (colorectal) cancer tissue[23]. A set of immunoglobulin genes show *cis*-eQTL effects that are only present under inflammatory conditions (linear regression, $t$ test, $FDR_{interaction}$ < 0.05). These genes (*IGHV4-4*, *IGKV1-13*, *IGKV2-29*, *IGHV3-20*, *IGKV1D-13*, *IGKV1D-27*, *IGKV1D-17*) are all involved in antigen recognition by B cells. Interestingly, the eSNP rs11685391 affects the expression of three of these genes (*IGKV1D-27*, *IGKV1D-17*, and *IGKV2-29*), suggesting that there is common genetic regulation (Fig. 1b).

Twenty-four out of 190 *cis*-eQTLs are dependent on eSNP genotype and also interact with inflammation. For example, a *cis*-eQTL effect between rs6860770 (A/C, linear regression, $t$ test, $FDR_{genotype}$ = 2.26e − 16) and gene *C6*, a complement protein encoding gene, which plays a role in the innate and adaptive immune response[24], is observed in the whole sample. However, the effect size is different in inflamed tissue compared with non-inflamed tissue (linear regression, $t$ test, $FDR_{interaction}$ = 0.014; Fig. 1c).

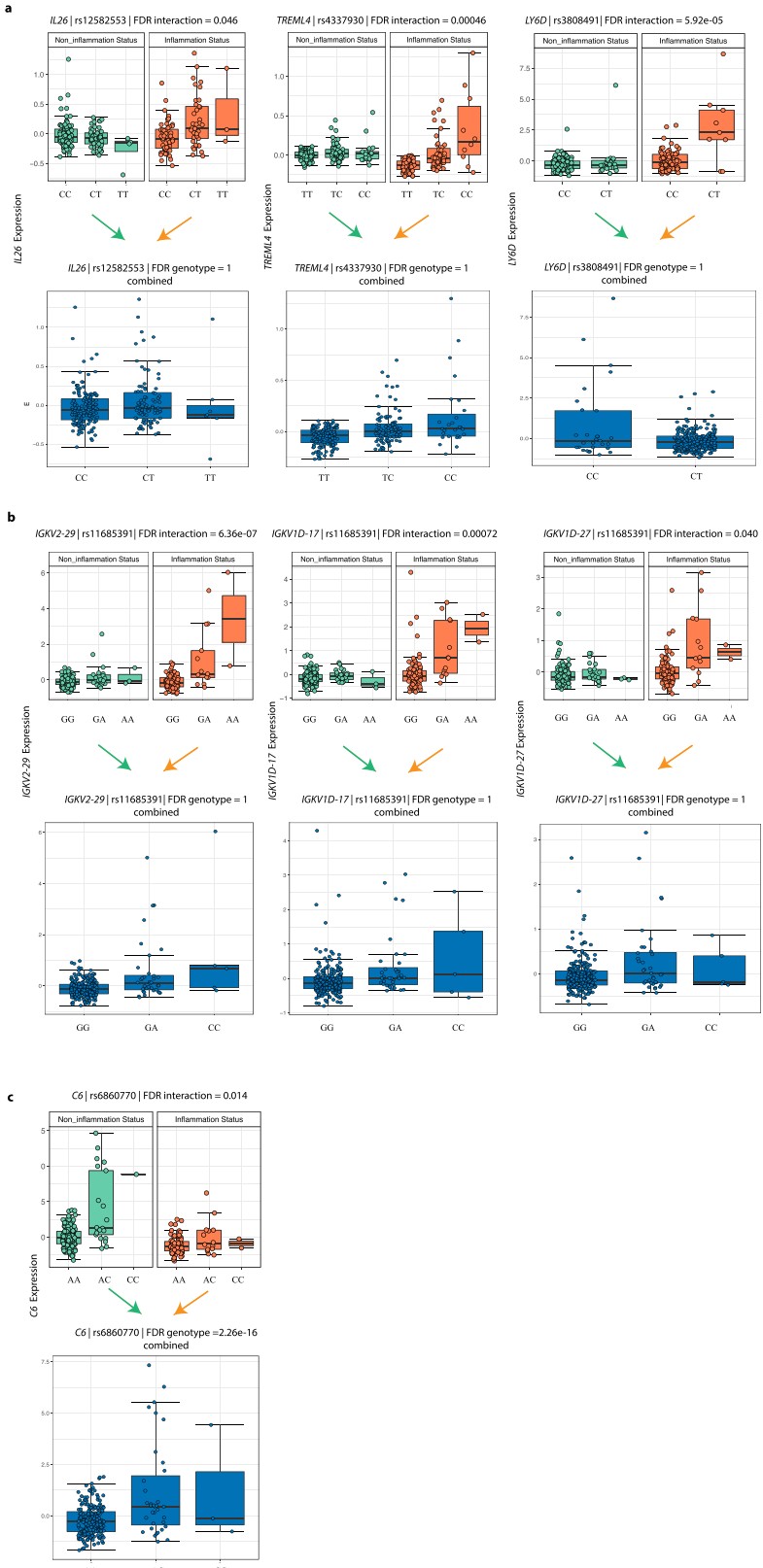

Pathway analysis of the 190 inflammation-dependent *cis*-eQTL genes revealed top pathways related to membrane protein anchoring ("posttranslational modification: synthesis of glycosylphosphatidylinositol-anchored proteins," $P = 6.95e − 06$), synthesis of inflammation-signaling proteins ("synthesis of leukotrienes and eoxins," $P = 2.44e − 04$) and bacterial recognition ("uptake and actions of bacterial toxins," $P = 0.0015$) (Supplementary Data 7). These are all factors that can influence an individual patient's pathophysiology. Other pathways indicate involvement of adipocyte differentiation ("transcriptional regulation of white

**Fig. 1 Inflammation-dependent *cis*-eQTLs. a** Three examples of *cis*-eQTLs only driven by interaction, rs12582553-*IL26* (linear regression, *t* test, FDR$_{interaction}$ = 0.046), rs337930-*TRENL4* (linear regression, *t* test, FDR$_{interaction}$ = 0.00046) and rs3808491-*LY6D* (linear regression, *t* test, FDR$_{interaction}$ = 5.92e − 05). Top panel: *X*-axis indicates the genotypes of the eSNPs stratified by inflammation status. *Y*-axis indicates the scaled expression levels of the eGenes. Bottom panel: *X*-axis indicates the genotypes of the eSNPs in all samples combined. *Y*-axis indicates scaled expression levels of the eGenes. **b** Three inflammation-dependent immunoglobulin *cis*-eQTLs. Panels are similar to (**a**). **c** One example of *cis*-eQTL driven by both genotype and the interaction, rs6860700-*C6* (linear regression, *t* test, FDR$_{interaction}$ = 0.014; FDR$_{genotype}$ < 2.26e − 16). Box plots show medians and the first and third quartiles (the 25th and 75th percentiles), respectively. The upper and lower whiskers extend the largest and smallest value no further than 1.5 × IQR (*n* = 280 samples). Source data are provided as a Source Data file.

adipocyte differentiation," $P = 1.94e − 04$) and phagocytosis ("FGFR2 ligand binding and activation," $P = 2.49e − 04$; "Fcgamma receptor-dependent phagocytosis," $P = 0.0042$).

To get a broader functional explanation of the inflammation-dependent *cis*-eQTL genes, we explored whole-transcriptome-wide co-expression patterns for each of the 190 eGenes and found 2466 co-expressed genes (Spearman's correlation, $|r| > 0.5$, FDR < 0.05) (Supplementary Data 8). For example, *IL26* is co-expressed with the IBD-associated gene *GPR25* ($r = 0.51$). *CHL1-AS2*, a noncoding RNA regulated by eSNP rs11685391, is co-expressed with the gene *CLEC12B* (Spearman's correlation, $r = 0.51$), a pathogen-recognition molecule in mucosal macrophages. In addition, *IL36RN* is co-expressed with *SPRR3* (Spearman's correlation, $r = 0.51$), together with a set of small proline-rich region genes that include *SPRR2D*, *SPRR2A*, and *SPRR1B*. These genes have been reported to be part of the inflammatory response in the epithelial barrier[25,26]. These results indicate that inflammation-dependent *cis*-eQTLs are potentially involved in gene–gene interactions.

**Cell types likely perturbed by inflammation *cis*-eQTLs.** Since cell type composition can change depending on inflammation status[27] and genotype[28] and eQTLs may be dependent on cell type composition[29], we investigated the contribution of the observed eSNPs to cell type heterogeneity.

To identify which of the 190 inflammation-dependent *cis*-eQTLs are related to the enrichment of (one of) the 28 intestinal cell types, we used xCell, an R package that works with a large reference base of 1182 transcriptomes[11]. Within xCell, cell type-enrichment scores can be calculated for 28 cell types present in the intestine. In short, we built interaction models including the genotype of *cis*-eQTL, deconvoluted cell type enrichment and calculated the interaction between these two ("Methods"). Each of these 28 cell type-enrichment scores showed a significant interaction (linear regression, *t* test, FDR$_{interaction}$ < 0.05) with one or more inflammation-dependent eSNPs. One hundred and twenty-five of the 190 inflammation-dependent *cis*-eQTLs show a significant interaction (linear regression, *t* test, FDR$_{interaction}$ < 0.05) with cell type enrichment (Fig. 2a and Supplementary Data 9). We identified significant *cis*-eQTL effects that are likely not only driven by genotype but also by a change in the frequency of specific cell types. For example, significant *cis*-eQTL effects were found between variant rs36065697 (G/A, linear regression, *t* test, FDR$_{genotype}$ <2.2e − 16) and gene *IGHV4-4*, variant rs76748970 (C/T, linear regression, *t* test, FDR$_{genotype}$ < 2.2e − 16) and gene *HLA-DQA2*, which also interact with an enrichment in macrophage M1 and plasma cells (Fig. 2b). Ninety-six out of the 125 eGenes only showed *cis*-eQTL effect with specific cell enrichment (linear regression, *t* test, FDR$_{interaction}$ < 0.05, FDR$_{genotype}$ > 0.05). For example, carriers of the TT and TA genotype of variant rs859739 upregulate *IGKV1D* gene expression with increasing of conventional dendritic cell (cDC) cell enrichment, while AA genotype carriers downregulate *IGKV1D* expression with decreasing cDC cells (linear regression, *t* test, FDR$_{interaction}$ = 6.65e − 06). However, there was no significant eQTL effect when merging the whole cDC cell populations (linear

regression, *t* test, FDR$_{genotype}$ = 0.11). This analysis demonstrates that inflammation-dependent *cis*-eQTLs may be partially driven by enrichment of specific cell types.

Of note, when comparing to the Westra study[30], where they searched for eQTLs that interacted with the proportions of blood cell subtypes, we find that the interaction effect of the eGene *TREML4* with eSNP rs4337930, in linkage equilibrium (LD) with rs6921835 ($r^2 = 0.97$), replicates in this dataset, indicating a tissue-overarching effect.

**Gene regulatory effects of eSNPs.** To further uncover the function of both the intestinal and inflammation-dependent *cis*-eQTLs, we set out to explore which eSNPs have a possible gene regulatory function. Using Haploreg annotation, we found that the top 200 intestinal eSNPs are enriched in enhancer regions in various gastrointestinal tissues, such as mucosa of the colon and the rectum ($P < 0.05$). For example, the eSNP rs2382817, a potential disease-dependent *cis*-eQTL with eGene *PNKD*, flanks an active transcription start site (TSS) in various gastrointestinal tissues. The 190 inflammation-dependent eSNPs show an enrichment for enhancer regions that have been found in blood immune cells, such as CD4 Th memory cells, monocytes, and B cells (Supplementary Data 10). Various inflammation-dependent eSNPs are located in active TSSs. For example, an inflammation-dependent eQTL for *AHSG*, rs3733160, flanks an annotated activated TSS in many tissues, including sigmoid, small intestine, and T and B cells from peripheral blood. The eGene *AHSG* is predicted to have a function in the pathway "GRB2: SOS provides linkage to MAPK signaling for Integrins," and rs3733160 might thus affect integrin signaling towards the intestine. rs144625530, an eSNP of the eGene *MLC1* that is involved in the "MAPK-erk pathway," is in LD with rs143407472 ($r^2 = 1$), which has active TSSs in primary Th memory cells from peripheral blood and colonic mucosa.

**Inflammation-dependent eGenes as possible drug targets.** The inflammation-dependent eQTLs are potentially involved in disease pathways, and targeting them could lead to a therapy that is highly specific to both tissue and disease status, thereby limiting side effects. Therefore, we sought to identify currently available drugs that can target the eGenes. Using the OpenTargets database, we identified five inflammation-dependent eGenes that are drug targets. For example, *LAP3*, involved in protein turnover, is targeted by Tosedostat, which is currently used to treat acute myeloid leukemia[31]. Genes *CACNG7*, *CACNA2D3*, *RXRG*, and *SCN2A* are targets of various drugs (Supplementary Data 11).

**Discussion**

In the current study, we integrated genotype and gene expression data from the inflamed and non-inflamed intestinal mucosa of patients with IBD. We confirm that the biopsy location is the largest contributor (2145 differentially expressed genes) to differences in gene expression between samples[32,33], and show that inflammation is responsible for 1131 differentially expressed

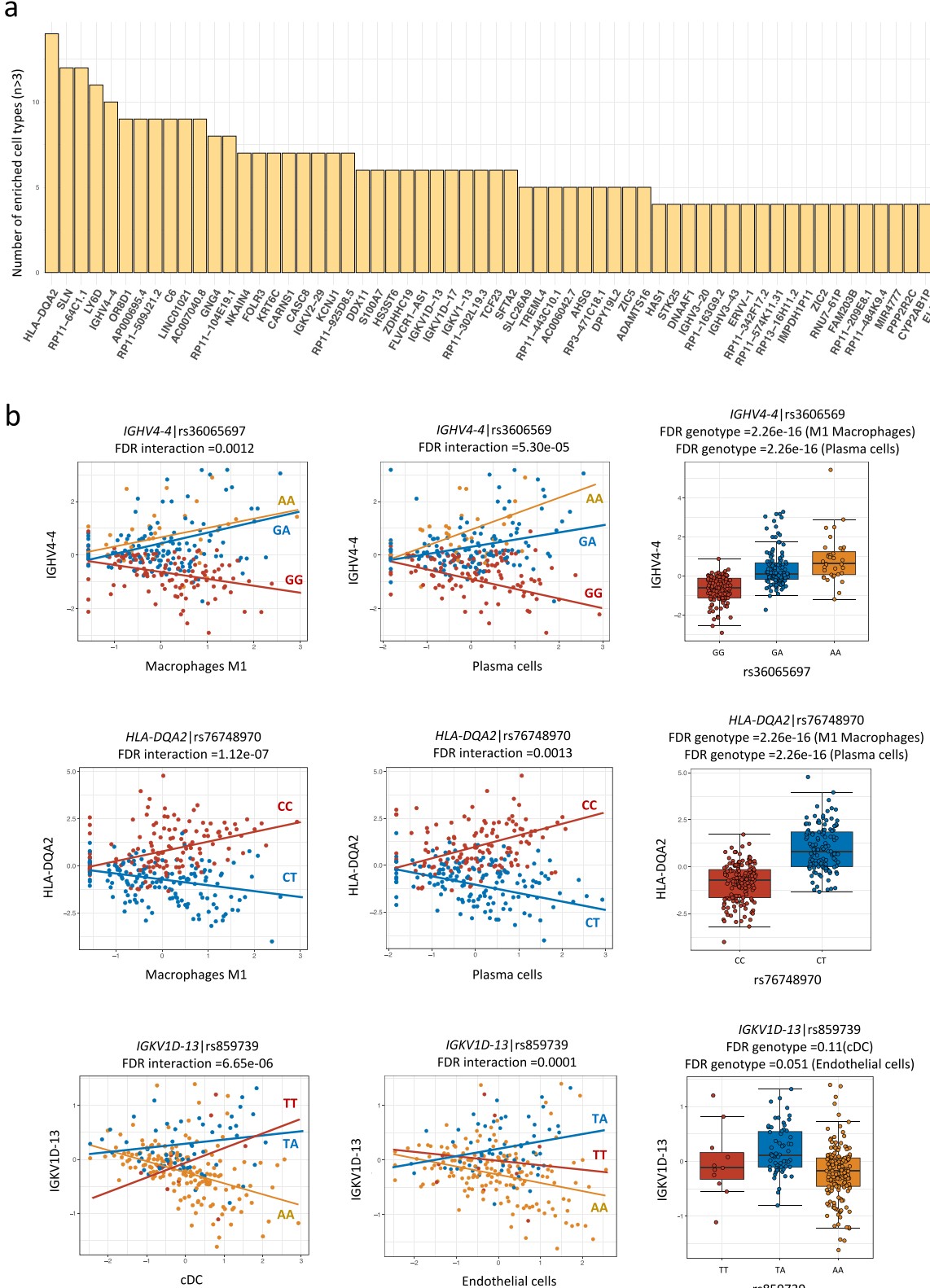

**Fig. 2 Associations between inflammation-dependent *cis*-eQTLs and cell type enrichment. a** One hundred and twenty-five out of 190 inflammation-dependent *cis*-eQTLs are associated with cell type enrichment. *X*-axis indicates gene IDs of *cis*-eQTLs. *Y*-axis indicates the number of associated enriched cell types (only *n* > 3 are shown). **b** Three examples. Left and middle panel, *X*-axis indicates cell type-enrichment scores derived from xCell. *Y*-axis indicates scaled expression levels of *cis*-eQTL genes. Colors indicate different genotypes. Right panel, *X*-axis indicates the genotype of *cis*-eQTL SNPs genotype, and *Y*-axis indicates scaled expression levels of *cis*-eQTL genes. Box plots show medians and the first and third quartiles (the 25th and 75th percentiles), respectively. The upper and lower whiskers extend the largest and smallest value no further than 1.5 × IQR (*n* = 280 samples). Source data are provided as a Source Data file).

genes. We identify 190 cis-eQTLs that depend on the inflammation status of the tissue. Downstream analyses of the genes we identified shows that inflammation-dependent cis-eQTL genes are involved in inflammatory responses to viruses and bacteria and mainly show interaction with certain subsets of immune cells. Overall, our results highlight the tissue and inflammation status specificity of cis-eQTLs.

We identified 8881 intestinal mucosa cis-eQTLs involved in general metabolic and transcriptional pathways. We also replicated 83.00% of the cis-eQTLs identified in the pediatric CD-based RISK cohort, indicating that a considerable number of the cis-eQTLs found in pediatric IBD may also be present in adult IBD. When comparing the intestinal cis-eQTLs with those identified in larger, non-disease specific datasets, including GTEx and CEDAR studies, we found an overlap of >92%, which supports the robustness of our findings. There are only a few studies on intestinal cis-eQTLs in IBD, and their findings are currently difficult to replicate because of differences in reporting[34]. Interestingly, four of the mucosal cis-eQTLs we identify (LIMD1, ZNF593, PPP2R2D, and RBL2) showed heterogeneity with inverse effect directions compared with the GTEx data, indicating that these four cis-eQTLs may be IBD-dependent. Furthermore, we showed colocalization of 330 intestinal eSNPs with genetic risk variants identified in GWAS of six gastrointestinal diseases (CD, UC, colon cancer, coeliac disease, and diverticulitis). We observed colocalization of eSNPs influencing the expression of HNF4A, ATG16L1, FUT2, and IRF8, which could be the causal variants of IBD, thereby linking GWAS findings to functional transcriptional effects in the intestinal mucosa.

Within the 190 inflammation-dependent eQTLs, we found a group of immunoglobulin heavy- and light-chain encoding eGenes (such as IGHV4-4). Immunoglobulins are resident in the human intestine, where they play an important role in maintaining the homeostasis between intestinal cells and the microbiome and other potentially harmful agents[35–39]. Our results suggest that immunoglobulin production may be disturbed in a genetically defined subset of patients, possibly resulting in different B cell-mediated IgA/IgG responses to intestinal bacteria and other triggers.

Expression of the antimicrobial cytokine IL26 is generally enhanced in the inflamed mucosa of patients with IBD, and it has been shown to induce pro-inflammatory cytokine expression in colonic subepithelial myofibroblasts[40,41]. We observe that IL26 expression in inflamed tissue depends upon genotype, suggesting that its antimicrobial defense mechanism might be more (or less) active in genetically defined subgroups of IBD patients. This could lead to variable susceptibility to superinfections, or simply to a difference in the duration of IBD flares. The interaction effect of eGene LY6D with inflammation status that we identified indicates a possible link between genotype, inflammation, and colorectal cancer, although this should be more thoroughly investigated in a dataset featuring both cancer and inflammation.

There have been earlier reports on inflammation-interaction cis-eQTLs in IBD tissue, which were found using source data of lower complexity[42]. However, we could not replicate any of the inflammation-interacting cis-eQTLs these authors reported, probably due to differences in sample origin, sample preparation, methods of data generation and analysis strategies.

A better understanding of how cell type composition differs under inflammatory conditions may provide targets for therapy[43,44]. Targeting gut mucosal- and inflammation-specific changes should render a therapy more specific, which would limit its side effects. Using cell type deconvolution of our bulk mRNA-seq data, we found a potential enrichment of various T cell subtypes in the inflammatory state, including Treg cells and Th2 cells. We also see an enrichment

in M1 (inflammatory) macrophages in inflamed tissue and of M2 macrophages in non-inflamed tissue, which is consistent with the literature[27,45]. We assessed the interaction between mucosal cell type-abundance and inflammation-dependent cis-eQTL effects to map possible confounding of cis-eQTL results caused by cell type composition and found that many inflammation-dependent cis-eQTLs may be linked to cell type abundance. Around one out of fourth of the interactions (112 out of 489) are with T cell subtypes, and 100 interactions are with (plasma) B cells. It is well established that T cells are highly present in the inflamed intestine, but interest in the role of B cells is growing, especially with respect to their interaction with the microbiome[46]. To definitively confirm these findings, absolute cell numbers would have to be determined, preferably accompanied by the cell type-specific expression patterns, for example, through single-cell RNA-seq.

We do find drug targets among inflammation-dependent cis-eQTLs, but not for drugs currently used to treat IBD. One explanation for this could be that the individuals included in this cohort use drugs that suppress their targets and therefore the possible eQTL effect. Another explanation may be that the eQTL effects of IBD drug targets are cell type-specific and cannot be found in whole biopsy RNA-seq data. It may also be that no IBD drugs currently target inflammation-dependent eQTLs.

The gene CXCL5 is an intestinal cis-eQTL. After treatment with the biological infliximab or vedolizumab, its expression was found to be downregulated in drug responders but not in non-responders[45], suggesting a possible genetic predisposition for drug response. Recently developed biological therapies are effective in roughly 30% of IBD patients[47], but we currently have no way to discern which patient will benefit from which therapy. Pharmacogenetics could be a way to predict drug effectiveness and toxicity prior to therapy[48]. Based on our cis-eQTL discoveries, we suggest drug–SNP combinations that could be interesting to investigate in large cohorts for genotype-dependent effectiveness.

Our study has some limitations. First, we show in silico that cis-eQTL effects can differ with inflammation status, but to assess causality, functional follow-up studies are needed. Second, our source data required correction for confounding factors such as biopsy location in order to calculate (inflammation-dependent) cis-eQTLs, which might have limited the number of results. Ideally, to minimize the amount of bioinformatic correction required, one would have paired biopsies of the same tissue, both inflamed and non-inflamed, from the same individuals. In practice, however, these samples are very difficult to obtain. Third, we assessed cis-eQTL effects per tissue and not per cell type[29]. By using single-cell mRNA transcriptomes for cis-eQTL analysis, one would be able to discriminate between a cell type-specific cis-eQTL and a cis-eQTL representing a cell type-enrichment effect. While we could estimate these effects using cell type deconvolution and interaction models, this approach cannot provide the same resolution as single-cell RNA-seq data[29]. Evaluating cis-eQTLs on the level of individual cells would be of great interest.

We have identified 190 inflammation-dependent and 8881 intestinal cis-eQTLs in IBD patients and highlight four potential IBD-specific intestinal cis-eQTL that, to our knowledge, have not been described to be associated with IBD. We also show which cell types likely contribute to these specific gene expression patterns and which regulatory features are potentially influenced by eSNPs. Our findings reveal that intestinal mucosa cis-eQTLs can depend on the inflammatory status and be influenced by the cell type composition of the mucosa. Our results highlight the importance of both genetics and the cell type composition of tissues as contributors to disease heterogeneity and provide leads that can guide drug development and to better profile patients for precision medicine.

## Methods

**Sample collection.** At the University Medical Center Groningen, a total of 299 intestinal mucosal biopsies were collected from 171 Dutch patients with a confirmed IBD diagnosis at the University Medical Center Groningen (Table 1). Biopsies were immediately snap-frozen on-site by the endoscopy nurse or research technician present during the endoscopy procedure. Information on biopsy location and macroscopic inflammation status was registered at the time of sample collection. Macroscopic inflammation status was classified based on the aspect of the mucosa during colonoscopy; inflamed defined as redness and edema with or without ulceration of the mucosa[49]. Blood for DNA extraction was drawn during prior visits to the hospital.

**Ethical approval.** All participants signed an informed consent form prior to sample collection. This study was approved by the Medical Ethics Committee of the University Medical Center Groningen (Groningen, The Netherlands, IRB ID 2008.338).

**Sample preparation.** Biopsies were homogenized in RLT plus buffer containing β-mercaptoethanol, using the Tissue Lyser with stainless-steel beads (Qiagen NV, Venlo, The Netherlands). Sample preparation was executed using the BioScientific NextFlex mRNA Sample Preparation Kit. DNA isolation was done with the AutoPure LS procedure from Qiagen. RNA was simultaneously isolated using the AllPrep RNA Mini kit (Qiagen), according to the manufacturer's protocol.

**Genotype data.** WES and GSA were performed on DNA derived from blood samples. WES data were obtained from 170 patients and generated at the Broad Institute of Harvard and MIT (Boston, USA), using an Illumina Hiseq 2500 sequencer. After quality control (QC) using default parameters (https://software.broadinstitute.org/gatk/best-practices/workflow?id=11146), reads were aligned to the b37 human reference genome, and subsequently, 86.06 million high-quality reads were generated per sample. On average, 98.85% of these reads aligned to the human genome (hg19) per sample. 81% of the whole exome reached an average >30× read depth and was used for further analysis.

GSA data were generated for all 171 IBD patients, using the Infinium GSA-24 v1.0 BeadChip combined with the optional Multi-Disease drop-in panel (http://glimdna.org/global-screening-array.html; GSA-MD). Genotypes were called using the OptiCall clustering program (ref. opticall.bitbucket.io), and QC steps were performed using PLINK 1.9 (www.cog-genomics.org/plink/1.9/) (minor allele frequency (MAF) > 5%, call rate < 0.99, Hardy–Weinberg equilibrium test $P$ value <1e − 4). Genotype data were phased using the Eagle algorithm and imputed to the Haplotype Reference Consortium reference panel using the Michigan Imputation Server (https://imputationserver.readthedocs.io/en/latest/pipeline/). After imputation, genetic variants were filtered for $R^2 > 0.4$ and MAF > 0.1%. For one sample only, GSA rather than WES was available.

GSA genotype data was combined with WES data using PLINK 1.9, and genomic variant filtering was performed. Variants with a call rate <0.99, a MAF <5%, and Hardy–Weinberg equilibrium test $P$ value < 1e − 6 were removed. The resulting filtered, combined WES-GSA genetic dataset covered a total of 6,894,979 genomic variants. The kinship matrix was calculated using PLINK 1.9 with parameters "--distance ibs" for all samples.

**Transcription data.** Paired-end RNA-seq was performed on all 299 biopsy samples using the Illumina NextSeq500 sequencer (Illumina). The RNA samples were pseudo-randomized on plates to mitigate batch effects covering IBD diagnosis, disease location, and disease activity. Twenty million paired-end 75-bp reads were generated per sample. The quality of the raw reads was checked using FastQC with default parameters (v0.11.7). The adaptors identified by FastQC were clipped using Cutadapt (v1.1) with default settings. Sickle (v1.200) was used to trim low-quality ends from the reads (length <25 nucleotides, quality <20). Reads were aligned to the human genome (human_g1k_v37) using HISAT (v0.1.6) (two mismatches allowed), and read sorting was done using SAMtools (v0.1.19). SAMtools flagstat and Picard tools (v2.9.0) were used to obtain mapping statistics. Eight samples with a low percentage (<90%) of read alignment were removed. Finally, the gene expression was estimated through HTSeq (0.9.1) based on the Ensemble version 75 annotation, resulting in an RNA expression dataset of 28,746 genes. Gene-level expression data were normalized using a trimmed mean of $M$ values, and $\log_2$ normalization was applied.

**Principal component analysis and confounders.** Principal component analysis was performed on gene expression. Three samples were defined as outliers by visualization and removed. Two hundred and eighty samples were selected for further analysis (Table 1).

To account for as many potential confounders as possible in our analysis, we chose the inflection of the variance curve[9,50], (Supplementary Figure 5) as the PC-correction threshold. The inflection occurs after the first 18 PCs, which together explain 77% of the total variation. Location of the biopsy (ileum vs. colon) was captured by the first PC (Spearman's correlation, $r^2 = 0.64$, $P = 3.40e − 66$), and inflammation status was captured by the second PC (Spearman's correlation, $r^2 = 0.41$, $P = 2.64e − 34$). Using this method, other potential confounding factors were visualized and corrected for (Supplementary Fig. 6A, B).

**Differential gene expression analysis.** To identify genes that were differentially expressed between disease groups (CD vs. UC), inflammation status groups (inflamed vs. non-inflamed), and biopsy location groups (ileum vs. colon) (Table 1), we applied a linear mixed mode using GEMMA[51] with identity-by-state matrix as a random effect for controlling repeated measurements and genetic relatedness, and "group" as a fixed effect. The first PC was excluded as a covariate when comparing disease and location groups. The second PC was excluded as a covariate when comparing inflammation status. By doing so, we corrected for PCs that account for other confounders. Our model can be described by the following formula:

$$\text{differential gene expression between inflammation status} \\ = (\text{intercept}) + \text{PCs}(1 + 3 \sim 18) + \text{inflammation status} + \text{IBS matrix} \quad (1)$$

$$\text{differential gene expression between diseases or sample locations} \\ = (\text{intercept}) + \text{PCs}(2 \sim 18) + \text{disease/location} + \text{IBS matrix} \quad (2)$$

Twenty-eight IBD unclassified samples were grouped with UC samples for further analyses.

**Cis-eQTL analysis.** For transcriptome-wide *cis*-eQTL mapping, we included SNPs located within 500 kb of a gene center, based on Ensemble v.75 annotation. The best guess of genotypes of the SNPs was used and encoded as 0, 1, or 2 to represent the number of the three genotypes. A linear mixed model was applied using GEMMA to identify intestinal *cis*-eQTLs. We corrected for confounders by regressing out the effect of the first 18 PCs. Our model can be described by the following formula:

$$\text{gene expression for } \textit{cis-}\text{eQTL analyses} = (\text{intercept}) + \text{PCs}(1 \sim 18) + \text{SNP} + \text{IBS matirx} \quad (3)$$

*Comparison to other* cis-*eQTL data.* To assess the robustness of our approach and dataset, significant *cis*-eQTLs (FDR < 0.05) were aligned to three publicly available datasets: (1) GTEx significant *cis*-eQTL ($q$ value < 0.05) summary statistics[12] for Colon_sigmoid ($n = 124$), Colon_transverse ($n = 169$), and Small_intestine ($n = 77$), 2) significant intestinal eQTLs (FDR < 0.05) of the "CEDAR" study[5] ($n = 323$), including ileum, transverse colon and rectum, (3) significant eQTLGen[14] blood eQTLs (FDR < 0.05) from 37 population-based cohorts ($n = 31,684$) (https://www.eqtlgen.org/), and (4) the pediatric IBD "RISK" cohort results[13] ($n = 245$) with nominal $P$ value < 0.05. Proportional overlap was calculated as the proportion of significant *cis*-eQTLs deriving from our dataset, which replicated in these publicly available datasets with $\beta$ in the same direction. Heterogeneity test was performed using package "*metafor*"[52] in R (v.3.5.0). Furthermore, eSNPs (SNPs with a *cis*-eQTL effect) within IBD loci were compared to the results from the UC patients and familial adenomatous polyposis study of Kabakchiev and Silverberg[34] ($n = 173$).

**Inflammation-dependent *cis*-eQTL analysis.** To identify inflammation-dependent *cis*-eQTLs, we used gene–environment interaction function "-gxe" in GEMMA by adding an additional inflammation covariate to the model, and an interaction term between this covariate and genotype:

$$\text{gene expression for inflammation-dependent } \textit{cis-}\text{eQTL analysis} \\ = (\text{intercept}) + \text{PCs}(1, 3 \sim 18) + \text{SNP} + \text{inflammation} + \text{SNP} \quad (4) \\ \times \text{ inflammation} + \text{IBS matrix}$$

We calculated $P$ values for the interaction terms and applied FDR correction. However, only $\beta$, standard errors, and $P$ values of the interaction term were obtained from GEMMA by default. To get the full summary statistics of the whole model, we re-calculated the significant inflammation-dependent *cis*-eQTLs using "*lme4qtl*" R package[53]. The FDR was calculated for the $P$ values of coefficients for variable genotype, inflammation, and the interaction terms separately. The significant threshold was $\text{FDR}_{\text{interaction}} < 0.05$.

**Colocalization of *cis*-eQTL SNPs with diseases GWAS.** We extracted all variants that were used for each significant intestinal *cis*-eQTL gene and performed colocalization analysis using *coloc* (v.3.2)[15] R package. GWAS summary statistics of six diseases were downloaded from https://www.ebi.ac.uk/gwas/, including IBD (ebi-a-GCST004131), CD (ebi-a-GCST004132), UC (ebi-a-GCST004133), coeliac disease (ukb-b-8631), diverticulitis (ukb-b-14796), and colon cancer (ukb-b-20145). For each test, we used the posterior probability of a model with one common causal variant (PP4) > 0.5 as colocalization evidence between eSNP and GWAS variants.

**Co-expression analysis.** Transcriptome-wide co-expression analysis was done for all inflammation-dependent *cis*-eQTL genes after adjusting for PCs. Co-expressed genes with FDR < 0.05 and the absolute value of Spearman's correlation coefficient >0.5 were selected.

$$\text{inflammation-dependent } \textit{cis-}\text{eQTL gene} = \text{PCs}(1 \sim 18) + \text{gene} + \text{IBS matrix} \quad (5)$$

**Cell type-specific analysis.** To order to define the cell types most likely to be driving or perturbed by inflammation-dependent *cis*-eQTLs, we calculated the

interaction between the *cis*-eQTL effect and a cell type-enrichment score. We used the xCell package (https://github.com/dviraran/xCell) for cell type-enrichment analyses, stratifying the following cell types that can be present in the intestinal mucosa: cDCs, pDCs, M1 Macrophages, M2 Macrophages, NK cells, naive CD4+ T cells, central memory CD4+ T cells, effector memory CD4+ cells, naive CD8+ T cells, central memory CD8+ T cells, effector memory CD8+ cells, Tyd cells, Th1 cells, Th2 cells, Tregs, NKT cells, naive B cells, plasma cells, memory B cells, class-switched memory B cells, basophils, MAST cells, neutrophils, eosinophils, endothelial cells, epithelial cells, fibroblasts and smooth muscle cells. The healthy cell type marker genes as published by Smillie et al.[27] were used to assess cell-marker enrichment.

$$
\begin{aligned}
\text{inflammation-dependent } cis\text{-eQTL gene} \\
= (\text{intercept}) + \text{PCs}(1 \sim 18) + \text{SNP} + \text{cell type-enrichment score} + \text{SNP} \quad (6) \\
\times \text{ cell type-enrichment score} + \text{IBS matrix}
\end{aligned}
$$

We calculated *P* values for the genotype, cell type-enrichment score, and the interaction terms using "lme4qtl" R package[53] and applied FDR correction separately. An interaction between cell type enrichment and a *cis*-eQTL was considered significant if the $\text{FDR}_{\text{interaction}} < 0.05$. We compared our results to those of the Westra et al. study, which uncovered eQTL–cell type-proportion interactions in blood[30]. Results were considered overlapping when the eQTL–cell type associations were significant in both cohorts.

**Functional annotation**. We performed a series of downstream analyses to further characterize the inflammation-dependent *cis*-eQTLs we had identified.

*Pathway analyses*. We used GeneNetwork (https://genenetwork.nl/) to provide functional enrichment analyses and report the top-20 upregulated pathways.

*Drug–target comparison*. To assess whether the *cis*-eQTL genes we identified are potential drug targets, we overlapped *cis*-eQTL genes with DrugBank and Open-Targets (https://www.drugbank.ca/, www.opentargets.org).

*Regulatory feature annotation*. We used Haploreg v4.1 (https://pubs.broadinstitute.org/mammals/haploreg/haploreg.php) to annotate regulatory features of eSNPs, and to calculate the number of SNPs that can be analyzed is limited, we restricted the number of SNPs used for the enrichment calculations to the top 200.

**Statistics**. All results were corrected for multiple testing using the "p.adjust" function (FDR, Benjamini–Hochberg method) in R (v.3.5.0), hereafter called "FDR." Because we did not have a replication set for the inflammation-interaction *cis*-eQTL analysis, we validated the FDR threshold by randomly assigning the inflammation status ten times and performing transcriptome-wide inflammation-dependent *cis*-eQTL analyses. For each permutation, we determined the interaction term *P* values, applied the FDR, and determined the number of interaction effects with an FDR < 0.05. If this number was <0.05× the number of interaction results, the FDR threshold used was considered adequate.

**Reporting summary**. Further information on research design is available in the Nature Research Reporting Summary linked to this article.

## Data availability

The raw gene expression table and full eQTL summaries data generated in this study have been deposited in the Genome-phenome Archive data repository database under accession code EGAS00001002702 ("Multi-omics data of 1000 Inflammatory Bowel Disease patients," datasets numbers: EGAD00001006789, EGAD00001006790, EGAD00001006791, EGAD00001006792, and EGAD00001006798). Due to participant confidentiality, the raw sequencing data and clinical phenotype data are available upon request of a letter of intent to the 1000IBD Data Access Committee UMCG. The publicly available datasets used in this study include: six diseases GWAS summary statistics were downloaded from https://www.ebi.ac.uk/gwas/, including IBD (ebi-a-GCST004131), CD (ebi-a-GCST004132), UC (ebi-a-GCST004133), coeliac disease (ukb-b-8631), diverticulitis (ukb-b-14796), and colon cancer (ukb-b-20145); GTEx significant *cis*-eQTL summary statistics were derived from (https://gtexportal.org/home/datasets, GTEx_Analysis_v7_eQTL.tar.gz); the "CEDAR" study intestinal eQTLs were downloaded from https://www.nature.com/articles/s41467-018-04365-8; eQTLGen blood eQTLs are obtained from https://www.eqtlgen.org/; the pediatric IBD "RISK" cohort eQTLs results are downloaded from https://www.nature.com/articles/ng.3936. The remaining data are available within the Article or from the authors upon request. Source data are provided with this paper.

## Code availability

Codes used for the following data processing and analysis are publicly available at: https://github.com/WeersmaLabIBD/RNA-SEQ (https://doi.org/10.5281/zenodo.4304528).

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

## Acknowledgements
We thank the patients of the 1000IBD cohort for contributing blood and intestinal biopsies. Many thanks to the doctors, nurses, and technicians for arranging and processing the samples. We also thank Kate Mc Intyre for English and content editing.

## Author contributions
S.H. and W.T.U.V. contributed to analysis and writing of the manuscript; H.-J.W., A.V.V. and R.B. contributed to the scientific discussion concerning the analysis and results; M.D.V. and Y.L. processed GSA data; T.B. and B.H.J. processed all samples in the lab; M.J.D. and R.J.X. provided WES data; E.A.F., G.D. and R.K.W. provided clinical samples and supervised analyses and the scientific discussion.

## Competing interests
The authors declare no competing interests.
