## [Peer Review File · Nature Communications]

REVIEWER COMMENTS

Reviewer #1 (Remarks to the Author):

The authors have generated transcriptomic data on a large collection of intestinal biopsies from inflammatory bowel disease patients, including samples from inflamed and non-inflamed tissue, from different locations and from different diagnoses (CD vs UC), along with matched exome sequencing and whole-genome genotyping.

As a resource alone, this is highly valuable to the research community that focuses on the genetics of intestinal diseases, and fills a gap that has to date consisted largely of array studies or studies of healthy individuals. It will also be a valuable resource to help interpreting and sense-checking the coming single-cell RNA-seq data that is in the pipeline in various countries.

The paper describes its methods well (modulo some comments below), uses a range of bioinformatic analyses (particularly focused on pathway analyses) to help interpret the data, and compares to external datasets (including both other transcriptomic datasets and external GWAS results). Some of these analyses could benefit from some more detail, in particular some more clarity on what novel biological lessons we can learn from this dataset (again, see comments below). There is some discussion of, e.g., what these results can tell us about drug targets, which I think again would benefit from some additional clarity.

Major point 1:

I think that this paper is probably best understood as presenting a valuable community resource first and foremost, with the direct scientific findings being of secondary value. As a result, ensuring that the data provided is of maximum value to the scientific community is key to the value of this paper. As far as I could see from the paper, the only data that is made directly available from this analysis is the list of hits SNPs from 122 inflammation-dependent eQTLs.

At the very least, links to download the full summary statistics for every SNP/gene pair tested should be included. Ideally, quantified gene expression data should be included in an open database such as ArrayExpress or GEO, and genotype and sequencing data should be made available using a controlled access platform such as EGA or dbGaP. This would dramatically increase the value of this dataset. The GTex study is a good model for how this data can be made available (and its very broad use is a testament to the value of doing so). The Nature Research data sharing policy ("Reporting standards and availability of data, materials, code and protocols") includes helpful information about the preferred modes of data sharing for articles published in their journals.

If any of this is already available, it needs to be clearly signposted in the text of the paper.

Major point 2: Differences in direction of effect

The direction of effect changes between GTex and the authors data (line 135 onwards) are interesting, but they are hard to interpret and seem like they could be prone to false positives. Looking up the GTex associations for these variants on the GTex and OpenTargets browsers I was only able to replicate one of them (RMI2 in sigmoid colon - the others did not show up as significant eQTLs), which suggests that they may be either marginal or unstable associations in GTex. It would be helpful if the authors could carry out a formal test of heterogeneity at these variants using the effect size and standard errors from the two datasets.

Supplementary table 5d does not give a lot of information given for these associations, and more full summary statistics could help in assessing them. For instance, effect size, p-value, standard error and effect allele frequency for each tissue in GTex and Hu et al, along with formal heterogeneity statistics, would help, as would the summary statistics for inflamed and non-inflamed tissue and each of the locations.

Major point 3: IBD associations and colocalization

The eQTLs in IBD-associated regions (p5) are useful, and are a key output of this study. However, it is not clear from supplementary table 5 what the relationship between the eQTL and the actual IBD signal is. LD r^2 values with IBD hits are mentioned in the text, but full information on the IBD and eQTL lead variants in the tables, with r^2 between them and their effect sizes in both, would give more context. Ideally, the authors would run colocalization analyses to test whether the same causal variant is driving the eQTL and the IBD association.

I would also be interested in seeing whether other digestive-disease associated variants (e.g. coeliac disease, diverticulitis, colon cancer) show colocalizations with eQTLs in this dataset.

Major point 4: Celltype interactions and cell specificity

The interaction analysis with xCell-based celltype decomposition (p8) is a welcome part of this paper. However, I found it hard to interpret the results, both biologically and statistically.

I would like to see the full summary statistics for the model for these variants, i.e. the effect sizes, standard errors and p-values for all three of the parameters in the interaction model (i.e. the effect sizes for "SNP", "celltype enrichment score" and "SNP x celltype enrichment score", using the model laid out on line 517). I would also like to know the gene expression levels (or, equally, the effect size and significance of correlation between the celltype enrichment score and the gene of interest).

A cell type composition/SNP interaction on gene expression can occur (as noted in the main text) due to genotype effects on cell composition, cell-type specific eQTLs, or eQTLs that are present in one cell only after interacting with another cell. Finding out which of these is the case requires looking at the three parameters of the interaction model, as well as the gene expression level in the interacting cell type relative to average across the tissue, and in some cases using biological knowledge about the gene itself (e.g. the NKT cell interaction on the Ig gene can presumably only be explained by an NKT cell-B-cell interaction).

In summary, I would have like to have seen a bit more digging into these celltype interaction results to try and understand what is driving them.

Major point 5: Clarity around drug targets

The authors produce a list of drugs that target genes whose expression is differentially regulated by variants in inflamed vs non-inflamed samples (p9). I found this somewhat difficult to interpret biologically - why would we be particularly interested in drugs that target this set of genes (more than, e.g., the large number of genes that are eQTLs in both inflamed and non-inflamed tissue, or genes that are expressed higher in inflamed tissue)? Some more detail in the results or discussion

walking the reader through why these eGenes are of particular value would help.

Some of these drugs appear rather broad in their mechanisms - for instance, acitretin targets a range of retinoid receptors. It might be worth highlighting in supplementary table 12 which drugs specifically target the specified protein, vs those that target a protein family or general pathway.

The inflammation-dependent eQTL in CXCL5 (line is interesting. Given the existence of genotyped datasets that have studied response to infliximab, e.g. the PANTS dataset, the authors could presumably test the hypothesis that this variant is expected to correlate with treatment outcome.

Minor point 1: Methodological details

I had a number of questions about the analysis, many of which I eventually found answered in the "Analysis.pipeline.md" file in the GitHub link included in the "Materials & Correspondence" at the end of the paper. Please include a link to this codebase near the start of the methods section. It would also be useful to clarify what the relationship is between that codebase and the other GitHub linked in the methods (eQTL-mapping-analysis-cookbook-for-438RNA-seq-data).

It would be useful for the authors to review the pipeline readme and the analysis, and ensure that all detail included is at least alluded to in the paper. This is particularly true for the use of GEMMA, which I found hard to reconstruct from the text. A few specific comments on that point:

- The calculation (and indeed use) of the kinship matrix does not seem to be discussed.
- Code for carrying out the differential expression analysis should be given - this is not in the GitHub pipeline provided. I do not actually know how the analysis described would be carried out in GEMMA and would not be able to repeat it.
- Some detail should be given on how repeated samples from the same individual are handled in the eQTL analysis - it isn't obvious to me how this is dealt with in the pipeline code.
- It might be helpful to state that the gene-environment interaction function in GEMMA is being used to test for inflammation-dependent eQTLs.
- The code uses some deprecated plink flags (e.g. --matrix, which should now be --distance ibs). Can the authors confirm that they used the stated version of plink (1.9) with deprecated flags, rather than using a deprecated version of plink. On a related note, the authors probably only need to give the plink URL the first time they mention the software, but it might help if they stated the plink build they used.
- The authors state that "SNPs were encoded as 0, 1, or 2 to represent the number of the three genotypes." -> does this mean that SNPs were coded according to the best guess genotype, rather than the imputed dosage?

Other minor points:

2. The authors rely on principal components to control for confounders such as disease location and inflammation status when carrying out the differential expression analysis (lines 458-459). It would be valuable to check that this is actually working as expected by also running analyses that explicitly condition on confounders (e.g. are CD vs UC differentially expressed genes significant after explicitly conditioning on inflammation and location). Obviously, this will restrict sample size (e.g. only colonic CD samples will be usable, assuming there are no UC ileal samples).

3. When pathway analyses from two different gene sets are compared (e.g. on lines 161-166) the authors should give the statistical significance on the size of the difference in enrichment (e.g. a heterogeneity test on the fold change). The top pathways that come up in an enrichment analysis can vary by chance, so a test to reject the null hypothesis that the same pathways are enriched in both datasets should be done.

4. For the gene-gene interaction analysis (line 198 onwards), if these co-expression effects are causal then the cis-eQTLs should generate trans-eQTLs on the correlated genes. E.g. if IL26 regulates GPR25, then IL26's cis-eQTL hit (rs125825530) should be correlated with GPR25 expressions. Did the authors test if this was the case, and compare the estimated trans-eQTL effect size to the effect size that would be expected based on the correlation?
5. I would be interested to know the overlap with the other major array-based eQTL study in intestinal biopsies (PMID:29930244), as well as overlap with whole blood eQTLs (which are often used as an eQTL source for IBD).

Reviewer #2 (Remarks to the Author):

Thank you for inviting me to review this interesting paper by Hu et al examining cis-eQTLs in IBD using fresh frozen samples from genotyped IBD patients. The paper is interesting as it identifies potential genes that may be involved in inflammation but suffers from the very heterogenous patient and sample groups and lack of discussion about these confounders.

The paper would be improved with the following:

1. Table 1: Was any statistical analysis done to determine if any significant differences between groups?
2. Table 2: Unclear why the SNPs were arranged seems to be random? ENS # - wouldn't p-value be more informative.
3. Samples inflamed vs non-inflamed: unclear which samples are from which patient - where these matched-pairs from the same patient? How do you define inflamed vs non-inflamed - pathologic scoring vs endoscopy? What is the definition of "lightly inflamed" - I have never heard of this term.
4. Patient - there are too many variables in this study - age, both CD and UC. Is there a possibility to use healthy controls? Not sure I understand how 80% of patients in the inflamed group are <40 years of age but the average age is 42? What is the age range of the other 20%.
5. Sample location and drug responses: how is disease location and multiple drug handled, drugs such as biologics must play a role in inflammation.
6. Functional analysis: there are a number of interesting genes that are superficially discussed. Some further functional studies of the candidates with greatly increase enthusiasm for this paper.

Reviewer 1:

The authors have generated transcriptomic data on a large collection of intestinal biopsies from inflammatory bowel disease patients, including samples from inflamed and non-inflamed tissue, from different locations and from different diagnoses (CD vs UC), along with matched exome sequencing and whole-genome genotyping.

As a resource alone, this is highly valuable to the research community that focuses on the genetics of intestinal diseases, and fills a gap that has to date consisted largely of array studies or studies of healthy individuals. It will also be a valuable resource to help interpreting and sense-checking the coming single-cell RNA-seq data that is in the pipeline in various countries.

The paper describes its methods well (modulo some comments below), uses a range of bioinformatic analyses (particularly focused on pathway analyses) to help interpret the data, and compares to external datasets (including both other transcriptomic datasets and external GWAS results). Some of these analyses could benefit from some more detail, in particular some more clarity on what novel biological lessons we can learn from this dataset (again, see comments below). There is some discussion of, e.g., what these results can tell us about drug targets, which I think again would benefit from some additional clarity.

Authors' response:

First of all, we would like to thank this reviewer for their kind words, the valuation of our work and for carefully reading our manuscript and providing valuable suggestions for improvement of our work.

Reviewer 1, Major point 1:

I think that this paper is probably best understood as presenting a valuable community resource first and foremost, with the direct scientific findings being of secondary value. As a result, ensuring that the data provided is of maximum value to the scientific community is key to the value of this paper. As far as I could see from the paper, the only data that is made directly available from this analysis is the list of hits SNPs from 122 inflammation-dependent eQTLs.

At the very least, links to download the full summary statistics for every SNP/gene pair tested should be included. Ideally, quantified gene expression data should be included in an open database such as ArrayExpress or GEO, and genotype and sequencing data should be made available using a controlled access platform such as EGA or dbGaP. This would dramatically increase the value of this dataset. The GTex study is a good model for how this data can be made available (and its very broad use is a testament to the value of doing so). The Nature Research data sharing policy ("Reporting standards and availability of data, materials, code and protocols") includes helpful information about the preferred modes of data sharing for articles published in their journals. If any of this is already available, it needs to be clearly signposted in the text of the paper.

Authors' response:

We would like to thank the reviewer for drawing attention to the crucial point of data sharing. When identified as a disease specific set from our center we cannot sufficiently pseudonymize our data, hence we are unable to put the quantified gene expression data in an open database by itself: However, links to download the full summary statistics for every SNP/gene pair tested and the quantified gene expression data will be made available through EGA (<https://ega-archive.org/studies/EGAS00001002702>, "Multi-omics data of 1000 Inflammatory Bowel Disease patients"), where it will be ready for upload once this manuscript is accepted.

Reviewer 1, Major point 2: Differences in direction of effect

The direction of effect changes between GTex and the authors data (line 135 onwards) are interesting, but they are hard to interpret and seem like they could be prone to false positives. Looking up the GTex associations for these variants on the GTex and OpenTargets browsers I was only able to replicate one of them (RMI2 in sigmoid colon - the others did not show up as significant eQTLs), which suggests that they may be either marginal or unstable associations in GTex. It would be helpful if the authors could carry out a formal test of heterogeneity at these variants using the effect size and standard errors from the two datasets.

Supplementary table 5d does not give a lot of information given for these associations, and more full summary statistics could help in assessing them. For instance, effect size, p-value, standard error and effect allele frequency for each tissue in GTex and Hu et al, along with formal heterogeneity statistics, would help, as would the summary statistics for inflamed and non-inflamed tissue and each of the locations.

Authors' response:

*We thank the reviewer for this constructive analysis. In the current version, we compared the 8,881 unique intestinal cis-eQTL pairs (FDR <0.05) from this study with significant cis-eQTLs ($qval \leq 0.05$) from three GTEx (v7) datasets: sigmoid, transverse colon and terminal ileum (<https://gtexportal.org/home/datasets>, GTEx_Analysis_v7_eQTL.tar.gz). We identified that six cis-eQTL pairs (rs8768 and ZNF593, rs4342945 and PPP2R2D, rs3736286 and ZFYVE19, rs10748 and RBL2, rs2742412 and LIMD1, rs13168973 and CCDC125) have inverse effect directions compared with all the three datasets. Then we extracted the six intestinal cis-eQTLs effect sizes and standard errors from this study and the GTEx (v7) and performed Cochran's Q-test using R package ("metafor"¹). A nominal P value <0.05 was considered significant and therefore, four pairs showed heterogeneity with GTEx, which indicates a potential disease effect on these cis-eQTLs. Full summary statistics including cis-eQTL effect size, standard error, allele frequency, P value and Q value of heterogeneity test is at **Supplementary Table 4b**. Plots of the these cis-eQTL effect are included in **Supplementary Figure 3** (GTEx eQTLs (v8) figures were obtained at <https://gtexportal.org/home/>, the pair between rs8768 and ZNF593 is not available online). The main text of the manuscript was adjusted:*

Results: line: 136-145

*"Interestingly, six eSNP-eGene have different directions of effect as compared to the GTEx study (data was not present at 'CEDAR' study). After a heterogeneity test between these eQTL pairs and the three GTEx gut datasets, four eQTL pairs showed significance, which suggests that these intestinal cis-eQTLs have indeed a different direction of effect in our dataset in in the context of IBD (Q test $P < 0.05$, **Supplementary Table 4b**, **Supplementary Figure 3**). These four eGenes consist of: PPP2R2D, a gene involved in the cell cycle by controlling mitosis entry and exit; RBL2, a gene associated with type 2 diabetes; LIMD1, a*

gene involved in several cellular processes including cell-cell adhesion and cell development and ZNF593, which modulates DNA binding. Neither the eGenes nor the eSNPs have been reported to be associated with IBD risk.”

Methods: line: 505

“Heterogeneity test was performed using package `metafor¹` in R (v.3.5.0).”

Discussion: line: 303-306

“Interestingly, four of these cis-eQTLs (LIMD1, ZNF593, PPP2R2D, and RBL2) showed heterogeneity with inverse effect directions compared with the GTEx data, indicating that these four cis-eQTLs may be IBD-dependent.”

Major point 3: IBD associations and colocalization

The eQTLs in IBD-associated regions (p5) are useful, and are a key output of this study. However, it is not clear from supplementary table 5 what the relationship between the eQTL and the actual IBD signal is. LD r^2 values with IBD hits are mentioned in the text, but full information on the IBD and eQTL lead variants in the tables, with r^2 between them and their effect sizes in both, would give more context. Ideally, the authors would run colocalization analyses to test whether the same causal variant is driving the eQTL and the IBD association.

I would also be interested in seeing whether other digestive-disease associated variants (e.g. coeliac disease, diverticulitis, colon cancer) show colocalizations with eQTLs in this dataset.

Authors' response:

We thank the reviewer for suggesting this analysis. In the previous version, we overlapped the significant intestinal cis-eQTL SNPs with all SNPs (LD $r^2 > 0.8$) at IBD GWAS loci². In the current version, to explore the relationship between the cis-eQTL SNPs and actual IBD GWAS signals, we extracted all variants for each of the 8,881 significant cis-eQTL genes and performed

colocalization analysis using the `coloc`³ R package. We downloaded six GWAS summary statistics from <https://www.ebi.ac.uk/gwas/>, including IBD (ebi-a-GCST004131), CD (ebi-a-GCST004132), UC (ebi-a-GCST004133), coeliac disease (ukb-b-8631), diverticulitis (ukb-b-14796) and colon cancer (ukb-b-20145). For each test, we considered a posterior probability of a model with one common causal variant (PP4) >0.5 as significant colocalization between cis-eQTL SNPs and diseases associated variants from GWAS. We found 172 cis-eQTL SNPs colocalized with IBD loci. For example, the most strongly colocalized cis-eQTL gene is HNF4A (PP4 =0.99). This gene was reported to have reduced expression in the intestinal mucosa in patients with IBD and UC compared to healthy people⁴. Four cis-eQTL SNPs colocalized with colon cancer and one with diverticulitis. 102 cis-eQTL SNPs colocalized with coeliac disease. Based on these findings, we extended the interpretation of the results and adjusted the main text in manuscript. The full summary statistics of colocalization analysis is at **Supplementary Table 5**:

Results: line: 148-165

Genomic variants within diseases susceptibility loci affect intestinal gene expression

To explore the functional impact of intestinal cis-eQTLs in gut diseases, we extracted GWAS summary statistics for six diseases traits, 1) IBD, 2) CD, 3) UC, 4) colon cancer, 5) diverticulitis, and 6) coeliac disease. Using `coloc`³, we performed colocalization analysis of the identified cis-eQTLs and disease GWAS loci to identify potential shared causal variants. At a posterior probability threshold of having one shared causal variant (PP4) of > 0.5, we discovered 558 colocalizing variants (**Supplementary Table 5**). For example, our IBD-based dataset showed 172 eSNPs that colocalized with IBD. The eGene that most strongly colocalized is HNF4A (PP4 =0.99), the expression of which is known to be decreased in the intestinal mucosa in patients with IBD and UC⁴. Functional enrichment analysis showed that the eGenes that colocalized with IBD GWAS loci are enriched for the 'Olfactory signaling pathway' (P value = 1.4e-08) and 'G alpha (s) signaling events' (P value = 1.6e-07). Both of these pathways are forms of G protein-coupled receptor signaling, which is a basic mechanism in the immune response in IBD⁵. For colon cancer, we found four colocalizing eSNPs and for diverticulitis we found one colocalizing eSNP. 102 eSNPs colocalize with coeliac disease, which are enriched for the 'ER-Phagosome pathway' (P value = 3.3e-06) and 'Nucleotide excision repair' (P value = 7.5e-06). ER stress pathways are known to play a

central role in IBD inflammation⁶. These results suggest that a large part of intestinal eSNPs are likely to be causal variants in IBD and coeliac disease.

Discussion: line: 306-308

Furthermore, we showed colocalization of 330 intestinal eSNPs with genetic risk variants identified in GWAS of six gastrointestinal diseases (CD, UC, colon cancer, coeliac disease, diverticulitis). We observed colocalization of eSNPs influencing the expression of HNF4A, ATG16L1, FUT2 and IRF8, which could be the causal variants of IBD, thereby linking GWAS findings to functional transcriptional effects in the intestinal mucosa.

Methods: line: 524-531

Colocalization of cis-eQTL SNPs with diseases GWAS

We extracted all variants that used for the each significant intestinal cis-eQTL gene and performed colocalization analysis using coloc (v.3.2)³ R package. Six diseases GWAS summary statistics were downloaded from <https://www.ebi.ac.uk/gwas/>, including IBD (ebi-a-GCST004131), CD (ebi-a-GCST004132), UC (ebi-a-GCST004133), coeliac disease (ukb-b-8631), diverticulitis (ukb-b-14796) and colon cancer (ukb-b-20145). For each test, we used the posterior probability of a model with one common causal variant (PP4) >0.5 as colocalization evidence between eSNP and GWAS variants.

Major point 4: Celltype interactions and cell specificity

The interaction analysis with xCell-based celltype decomposition (p8) is a welcome part of this paper. However, I found it hard to interpret the results, both biologically and statistically.

I would like to see the full summary statistics for the model for these variants, i.e. the effect sizes, standard errors and p-values for all three of the parameters in the interaction model (i.e. the effect sizes for "SNP", "celltype enrichment score" and "SNP x celltype enrichment score",

using the model laid out on line 517). I would also like to know the gene expression levels (or, equally, the effect size and significance of correlation between the celltype enrichment score and the gene of interest).

A cell type composition/SNP interaction on gene expression can occur (as noted in the main text) due to genotype effects on cell composition, cell-type specific eQTLs, or eQTLs that are present in one cell only after interacting with another cell. Finding out which of these is the case requires looking at the three parameters of the interaction model, as well as the gene expression level in the interacting cell type relative to average across the tissue, and in some cases using biological knowledge about the gene itself (e.g. the NKT cell interaction on the Ig gene can presumably only be explained by an NKT cell-B-cell interaction). In summary, I would have like to have seen a bit more digging into these celltype interaction results to try and understand what is driving them.

Authors' response:

*We fully agree with reviewer's suggestions that we should dig more into explanation of the cell type interaction models. Considering the P values of the interaction term, we identified that 125 out of the 190 cis-eQTLs show different effect sizes with different cell type enrichment ($FDR_{interaction} < 0.05$). We added the full summary statistics of cell type interaction models for these 125 cis-eQTLs, including effect sizes, P values and z scores for all the three terms (SNP, cell type enrichment score and interaction term between these two) (**Supplementary Table 9**). Considering the P values of the genotype term, 29 of the 125 are also likely driven by genotype independently, for example, variant rs36065697 (G/A) and gene IGHV4-4, showed both significance of the genotype ($FDR_{genotype} < 0.05$) and interaction with enrichment of macrophages M1 cells and plasma cells (**Figure 2B**). 96 out of the 125 showed cis-eQTL effects only after merging with specific cell enrichment ($FDR_{interaction} < 0.05$, $FDR_{genotype} > 0.05$). For example, carriers of the TT and TA genotype of variant rs859739 upregulate IGKV1D gene expression with increasing of cDC cell enrichment, while AA genotype carriers downregulate IGKV1D expression with decreasing cDC cells ($FDR_{interaction} = 6.65e-06$, **Figure 2B**). However, there was no significant eQTL effect when merging the whole cDC cell populations ($FDR_{genotype} = 0.11$). Based on the interaction model, it is difficult to interpret both the gene expression correlated with cell type enrichment and the cell-specific cis-eQTLs. Because correlated-genes/eQTLs identified this way are not necessarily specific to the estimated cell type*

enrichment but may reflect another correlated cell type (anti-correlated). Therefore, we would like to focus on the interpretation of the terms of the genotype and the interaction in the model and refer to these cis-eQTLs with $FDR_{interaction} < 0.05$ as cell type interaction cis-eQTLs. To confirm the genotype effects on cell type enrichment, absolute cell numbers would have to be determined, preferably accompanied by the cell type specific expression pattern, for example through single cell RNA sequencing. However, this interaction model could help to reveal cis-eQTL effects which are potentially obscured by cell type heterogeneity of biopsy bulk RNA-seq data. We adjusted the text in main manuscript:

Results: line: 231-250

we used xCell, an R package that works works with a large reference base of 1182 transcriptomes⁷. Within xCell, cell type enrichment scores can be calculated for 28 cell types present in intestine. In short, we built interaction models including the genotype of the cis-eQTLs, deconvoluted cell type enrichment and calculated the interaction between these two (Methods). Each of these 28 cell type-enrichment scores showed a significant interaction ($FDR_{interaction} < 0.05$) with one or more inflammation-dependent eSNPs. 125 of the 190 inflammation-dependent cis-eQTLs show a significant interaction ($FDR_{interaction} < 0.05$) with cell type-enrichment (Figure 2A, Supplementary Table 9). We identified significant cis-eQTL effects that are likely not only driven by genotype, but also by a change in frequency of specific cell types. For example, significant cis-eQTL effects were found between variant rs36065697 (G/A, $FDR_{genotype} < 2.2e-16$) and gene IGHV4-4, variant rs76748970 (C/T, $FDR_{genotype} < 2.2e-16$) and gene HLA-DQA2, which also interact with an enrichment in macrophages M1 cells and plasma cells (Figure 2B). 96 out of the 125 eGenes only showed eQTL effect with specific cell enrichment ($FDR_{interaction} < 0.05$, $FDR_{genotype} > 0.05$). For example, carriers of the TT and TA genotype of variant rs859739 upregulate IGKV1D gene expression with increasing of cDC cell enrichment, while AA genotype carriers downregulate IGKV1D expression with decreasing cDC cells ($FDR_{interaction} = 6.65e-06$). However, there was no significant eQTL effect when merging the whole cDC cell populations ($FDR_{genotype} = 0.11$). This analysis demonstrates that inflammation-dependent eQTLs may be partially driven by enrichment of specific cell types.

Discussion: line: 348-350:

To definitively confirm these findings, absolute cell numbers would have to be determined, preferably accompanied by the cell type specific expression patterns, for example through single cell RNA sequencing.

Discussion: line: 374-380:

Third, we assessed cis-eQTL effects per tissue and not per cell type⁸. By using single-cell mRNA transcriptomes for cis-eQTL analysis, one would be able to discriminate between a cell type-specific cis-eQTL and a cis-eQTL representing a cell type-enrichment effect. While we could estimate these effects using cell type deconvolution and interaction models, this approach cannot provide the same resolution as single cell RNA-seq data⁸. Evaluating cis-eQTLs on the level of individual cells would be of great interest.

Major point 5: Clarity around drug targets

The authors produce a list of drugs that target genes whose expression is differentially regulated by variants in inflamed vs non-inflamed samples (p9). I found this somewhat difficult to interpret biologically - why would we be particularly interested in drugs that target this set of genes (more than, e.g., the large number of genes that are eQTLs in both inflamed and non-inflamed tissue, or genes that are expressed higher in inflamed tissue)? Some more detail in the results or discussion walking the reader through why these eGenes are of particular value would help.

Some of these drugs appear rather broad in their mechanisms - for instance, acitretin targets a range of retinoid receptors. It might be worth highlighting in supplementary table 12 which drugs specifically target the specified protein, vs those that target a protein family or general pathway.

The inflammation-dependent eQTL in CXCL5 (line is interesting. Given the existence of genotyped datasets that have studied response to infliximab, e.g. the PANTS dataset, the authors could presumably test the hypothesis that this variant is expected to correlate with treatment outcome.

Authors' response:

We thank reviewer for this suggestion. The inflammation-dependent eQTLs are potentially involved in disease pathways, and targeting them could lead to a therapy that is highly specific to both tissue and disease status, thereby limiting side-effects. Also it might be that eQTL effect that only become apparent during intestinal inflammation might be interesting targets in the

context of IBD. We also added two columns at **Supplementary Table 11**: one with the specificity, and one with a short summary of what is known on the working mechanism.

We retrieved the infliximab medication usage of the same patients with IBD (patients number =37, biopsy number =61) in this study from University Medical Center of Groningen (UMCG). Among them, 25 patients were responders to infliximab (defined as patients only treated with infliximab (≥ 1 year) without usage of a second biological, biopsy number =37), and 12 patients were non-responders (defined as patients who received vedolizumab or ustekinumab after infliximab treatment, biopsy number =24). We then performed regression analysis between cis-eQTL SNP rs187082 from gene CXCL5 and infliximab response groups. The best guess of genotype encoded as 0, 1 and 2 was used in a linear mixed model where the SNP was a fixed effect and the samples kinship as a random effect. However, we did not observe a significant association between this SNP and infliximab response (P value =0.866). We adjusted the text in the main manuscript:

Results: line: 275-282:

The inflammation-dependent eQTLs are potentially involved in disease pathways, and targeting them could lead to a therapy that is highly specific to both tissue and disease status, thereby limiting side-effects. Therefore, we sought to identify currently available drugs that can target the eGenes. Using the OpenTargets database, we identified five inflammation-dependent eGenes that are drug targets. For example, LAP3, involved in protein turnover, is targeted by Tosedostat, which is currently used to treat acute myeloid leukemia⁹. Genes CACNG7, CACNA2D3, RXRG and SCN2A are targets of various drugs (Supplementary table 11).

Minor point 1: Methodological details

I had a number of questions about the analysis, many of which I eventually found answered in the "Analysis.pipeline.md" file in the GitHub link included in the "Materials & Correspondence" at the end of the paper. Please include a link to this codebase near the

start of the methods section. It would also be useful to clarify what the relationship is between that codebase and the other GitHub linked in the methods (eQTL-mapping-analysis-cookbook-for-438RNA-seq-data).

Authors' response:

We thank the reviewer for this suggestion and it indeed clarifies the manuscript's methods sections. Now we added the GitHub link to the end of the Methods section (according to Nature Communications policies). We used the code for gene expression PCA calculation from "eQTL-mapping-analysis-cookbook-for-438RNA-seq-data" in the old GitHub. We re-calculated the PCA in R ('prcomp' function) and clarified it in the current version of GitHub (<https://github.com/WeersmaLabIBD/RNA-SEQ/blob/master/Analysis.pipeline.md>, section "Remove PCs"). The "eQTL-mapping-analysis-cookbook-for-438RNA-seq-data" was removed.

Methods: line: 586-588

Code availability

Codes used for the following data processing and analysis are publicly available at:

[\[https://github.com/WeersmaLabIBD/RNA-SEQ\]](https://github.com/WeersmaLabIBD/RNA-SEQ).

It would be useful for the authors to review the pipeline readme and the analysis, and ensure that all detail included is at least alluded to in the paper. This is particularly true for the use of GEMMA, which I found hard to reconstruct from the text. A few specific comments on that point:

- The calculation (and indeed use) of the kinship matrix does not seem to be discussed.

Authors' response:

We thank the reviewer for pointing out the missing information on the kinship matrix calculation. Now we added it into the manuscript:

Methods: line: 437-438

“The kinship matrix was calculated using PLINK 1.9 with parameters `--distance ibs` for all samples.”

- Code for carrying out the differential expression analysis should be given - this is not in the GitHub pipeline provided. I do not actually know how the analysis described would be carried out in GEMMA and would not be able to repeat it.

Authors' response:

We fully agree that we should provide detailed information on differential gene expression calculation. GEMMA requires bimbam files which contain genotype information to perform association between SNP and gene expression (from <https://github.com/genetics-statistics/GEMMA/blob/master/doc/manual.pdf>):

rs1, A, T, 0.02, 0.80, 1.50

rs2, G, C, 0.98, 0.04, 1.00

We replaced the genotype bimbam file with the phenotype file in GEMMA. For example, we coded sample inflammation as 1 and non-inflammation as 0, then we created a bimbam file containing inflammation instead of genotype:

rs00000 A T 1 1 0 1 1 1 1 0 1 1 0 1 0 1 1 1

where rs00000, A and T are random information while 1 and 0 indicates inflamed and non-inflamed samples. Then we used the following command:

```
gemma-0.98-linux-static -g Inflammation.bimbam -p gene.expression.txt -lmm 4 -km 1 -k IBS.mibs -o DGE.out
```

by which we could assess the association between inflammation and gene expression.

We updated the code at GitHub (<https://github.com/WeersmaLabIBD/RNA-SEQ/blob/master/Analysis.pipeline.md>, section “Part 1. differential gene expression (DGE) analysis”)

- Some detail should be given on how repeated samples from the same individual are handled in the eQTL analysis - it isn't obvious to me how this is dealt with in the pipeline code.

Authors' response:

To control effect of repeated samples, we calculated a genetic kinship matrix (identity-by-state, IBS) using PLINK and used it as a random effect in linear mixed models in GEMMA. For example, the command for analyzing inflammation-dependent cis-eQTL was the following:

```
gemma-0.98-linux-static -bfile genotype.plink.file -p gene.expression.file -gxe covarite.file -km 1 -k IBS.mibs -lmm 4 -o outcome.file
```

Repeated samples are from the same individual and therefore, the IBS values between these samples are 1. In addition, we also compared models of differently handling the repeat measurements for the 190 significant inflammation-dependent cis-eQTLs pairs:

1) *Interaction model without adjusting for repeat measurements:*

*gene = intercept + PCs (1,3~18) + SNP + Inflammation + SNP*Inflammation*

2) *Interaction model using patients ID vector (1|ID) as a random effect:*

*gene = intercept + PCs (1,3~18) + SNP + Inflammation + SNP*Inflammation + (1|ID)*

3) *Interaction model using IBS matrix as a random effect:*

*gene = intercept + PCs (1,3~18) + SNP + Inflammation + SNP*Inflammation + (IBS matrix)*

We compared Z values of the interaction terms from the three models. A moderate difference between model 1) and 2) was observed (see figure below).

No difference between model 2) and 3) (see figure below).

We proved that in our eQTL analysis, it is reliable to use the IBS matrix as a random effect in linear mixed model to control for repeated measurements. We also updated the GitHub (<https://github.com/WeersmaLabIBD/RNA-SEQ/blob/master/Analysis.pipeline.md>, section “eQTL analysis – Run GEMMA” and <https://github.com/WeersmaLabIBD/RNA-SEQ/tree/master/lme4qtl>).

- It might be helpful to state that the gene-environment interaction function in GEMMA is being used to test for inflammation-dependent eQTLs.

Authors’ response:

We clarified that we used gene-environment interaction function of GEMMA to test inflammation-dependent cis-eQTLs in both the manuscript and the GitHub (<https://github.com/WeersmaLabIBD/RNA-SEQ/blob/master/Analysis.pipeline.md>, section "eQTL analysis – Run GEMMA").

Methods: line: 510-511

To identify transcriptome-wide cis-eQTLs that are inflammation-dependent, we used gene-environment interaction function `-gxe` in GEMMA by adding an additional inflammation covariate to the model, and an interaction term between this covariate and genotype.

- The code uses some deprecated plink flags (e.g. `--matrix`, which should now be `--distance ibs`). Can the authors confirm that they used the stated version of plink (1.9) with deprecated flags, rather than using a deprecated version of plink. On a related note, the authors probably only need to give the plink URL the first time they mention the software, but it might help if they stated the plink build they used.

Authors' response:

We thank the reviewer pointing out this issue. We indeed used deprecated plink flags `--cluster --matrix` to calculate IBS matrix with old version of PLINK. Now we updated the PLINK version to v1.90b3.32 and parameter to `--distance ibs`. We also added the PLINK URL to the first time mentioned and removed the rest in the manuscript.

- The authors state that "SNPs were encoded as 0, 1, or 2 to represent the number of the three genotypes." -> does this mean that SNPs were coded according to the best guess genotype, rather than the imputed dosage?

Authors' response:

Yes, we used the best guess genotype instead of imputed dosage of all the variants. We clarified it in the manuscript.

Methods: line: 487-488

The best guess of genotypes of the SNPs were used and encoded as 0, 1, or 2 to represent the three genotypes.

Other minor points:

2. The authors rely on principal components to control for confounders such as disease location and inflammation status when carrying out the differential expression analysis (lines 458-459). It would be valuable to check that this is actually working as expected by also running analyses that explicitly condition on confounders (e.g. are CD vs UC differentially expressed genes significant after explicitly conditioning on inflammation and location). – Obviously, this will restrict sample size (e.g. only colonic CD samples will be usable, assuming there are no UC ileal samples).

Authors' response:

*We agree with the reviewer suggestion, and performed differential expression analysis explicitly conditioning on tissue location, inflammation and disease diagnosis factors and compared the results with those based on principle components (PC) correction method. However, as the reviewer mentioned, splitting samples into different groups could lead to limited sample size. We restricted our comparison analysis using groups larger than 30 samples (See **table** below), therefore, UC ileum samples were not included in this analysis.*

Summary of sample numbers in different groups

	CD		UC	
	Colon	Ileum	Colon	Ileum
Inflamed	33	30	48	1
Non-inflamed	64	46	46	12

We first compared genes differently expressed between CD and UC in inflamed colon and non-inflamed colon samples separately. However, we didn't find significantly differentially expressed genes after multiple tests correction (FDR <0.05).

Comparison analysis CD vs. UC

$$gene = (intercept) + group(CD \text{ and } UC) + age + sex + batch + IBS \text{ matrix}$$

Second, we compared genes differentially expressed between location (colon/ileum) at CD inflamed and non-inflamed samples separately. Our results from the PCs-based correction largely overlap in this comparison analysis within CD samples (87.83%, same direction, 1884 out of 2145). The inconsistency between CD non-inflamed and CD inflamed samples could be caused by inflammation, power issues and other under-estimated confounders including medication and sequencing technique issues (See **Supplementary Figure 6**).

Comparison analysis colon vs. ileum

$$gene = (intercept) + group(colon \text{ and } ileum) + age + sex + batch + IBS \text{ matrix}$$

Third, we compared genes differentially expressed between inflamed and non-inflamed samples

at CD colon, CD ileum and UC colon separately. We also found our results from PC-based correction method have a large overlap with conditioning methods (71.00%, same direction, 803 out of 1131). The inconsistency between CD colon, CD ileum and UC colon could be driven by disease subtype, power issues and under estimated confounders (See **Supplementary Figure 6**).

Comparison analysis inflammation vs. non-inflammation

$$gene = (intercept) + group(inflamed\ and\ noninflamed) + age + sex + batch + IBS\ matrix$$

Although there are differences between the methods, we showed that our differently expressed genes from PC-based correction methods, at least for significant signals (FDR <0.05), are quite robust.

3. When pathway analyses from two different gene sets are compared (e.g. on lines 161-166) the authors should give the statistical significance on the size of the difference in enrichment (e.g. a heterogeneity test on the fold change). The top pathways that come up in an enrichment analysis can vary by chance, so a test to reject the null hypothesis that the same pathways are enriched in both datasets should be done.

Authors' response:

We thank the reviewer for this suggestion and we do agree that a heterogeneity test should be done when comparing pathway analysis in two gene sets. In the old version of the manuscript, we overlapped inflammation-dependent cis-eQTLs and intestinal cis-eQTLs. Then we performed pathway enrichment analysis for the non-overlapping eQTL genes (<https://www.genenetwork.nl/>, REACTOME enrichment). In the current manuscript, we explained the 190 inflammation-dependent cis-eQTLs by considering the significance of genotype term and the interaction term in a linear mixed model (**line in main text: 181-183; 198-203**). We found that 24 out of 190 cis-eQTLs are driven by both genotype effect and interaction effect. Pathway enrichment analysis does not add much biological value on the subset of cis-eQTLs and therefore, we removed it from the current manuscript.

4. For the gene-gene interaction analysis (line 198 onwards), if these co-expression effects are causal then the cis-eQTLs should generate trans-eQTLs on the correlated genes. E.g. if IL26 regulates GPR25, then IL26's cis-eQTL hit (rs125825530) should be correlated with GPR25 expressions. Did the authors test if this was the case, and compare the estimated trans-eQTL effect size to the effect size that would be expected based on the correlation?

Authors' response:

We do agree with the reviewer that if a cis-eQTL has a gene-gene interaction, this could be detectable as a trans-eQTL effect. However, detecting this trans-eQTL effect requires a lot of statistical power (i.e. a large cohort). Therefore, we performed a targeted genotype \times inflammation interaction analysis (the same model for inflammation-dependent cis-eQTL) to identify inflammation-dependent trans-eQTL within these 2,466 co-expressed genes. 4,470 trans-eQTLs were identified ($FDR_{interaction} < 0.05$, **table** below, top 20 listed). For example, the IL26's cis-eQTL SNP rs12582553 also has a regulation effect on gene GPR25 in context of inflammation ($FDR_{interaction} = 0.042$). The trans-eQTL effect is not all but one of the explanation for the genes co-expressed with cis-eQTL genes. For example, genes can be co-expressed with cis-eQTL genes due to pleiotropy effects or linkage disequilibrium of the SNPs^{10,11}. Therefore, we would like to focus on the cis-eQTL results in the main text without emphasizing trans-eQTLs.

Top 20 inflammation-dependent *trans*-eQTLs (ranked by $FDR_{interaction}$)

SNP	snp.Pvalue	snp.beta	inflammation.Pvalue	inflammation.beta	interaction.Pvalue	interaction.beta	interaction.z	interaction.FDR	Gene
rs11684047	0.107556757	-0.122858868	0	2.183273913	0	-1.061820524	-8.756958514	0	IGKVID-17
rs149065308	0.132307533	0.030977878	2.01E-12	0.541471376	2.22E-16	-0.326269555	-8.242491824	2.89E-12	KCNJ1
rs55680039	0.999445136	-1.18E-05	4.61E-09	0.333794392	3.13E-14	-0.223735713	-7.592428725	2.04E-10	RP11-256I23.2
rs149065308	0.870939058	-0.002702721	1.03E-08	0.356219951	2.99E-13	-0.233381675	-7.294678048	1.56E-09	OPRK1
rs111419523	0.100515735	0.022366683	1.27E-07	0.259405604	4.19E-13	-0.183415208	-7.249318014	1.82E-09	RN7SL76P
rs111419523	0.290146476	0.0197644	5.50E-09	0.382021984	5.78E-13	-0.243263744	-7.205533323	2.15E-09	RP11-375I20.6
rs111419523	0.064556843	0.023685278	2.10E-07	0.263332449	9.88E-13	-0.186526468	-7.132188662	3.22E-09	CTD-2620I22.3
rs10132224	0.957645076	0.000884214	8.57E-08	0.318540034	2.19E-12	-0.216016441	-7.022036859	6.33E-09	ATP5A1P5
rs149065308	0.497310793	0.01600755	9.23E-09	0.506708618	3.10E-12	-0.316276838	-6.972945852	8.09E-09	AC068657.2
rs859739	0.404691611	-0.033300538	2.89E-15	0.878408262	8.51E-12	-0.440045474	-6.829725569	2.02E-08	IGKV1-13
rs111419523	0.418171734	0.011912518	2.88E-07	0.287163663	1.32E-11	-0.195178148	-6.766922269	2.86E-08	RP11-793I11.1
rs149065308	0.520046442	0.010324316	4.77E-07	0.285745437	2.84E-11	-0.194226241	-6.6545385	5.70E-08	RP11-793I11.1
rs10132224	0.915577598	-0.001949076	2.90E-07	0.336418642	1.18E-10	-0.218476859	-6.442026151	2.20E-07	RP11-157P1.5
rs149065308	0.712674777	0.005141129	9.98E-06	0.230667235	3.36E-10	-0.168634492	-6.280993537	5.85E-07	CTD-2620I22.3
rs117682665	0.272365928	0.017724992	6.63E-06	0.233603972	5.00E-10	-0.169272651	-6.219068198	8.15E-07	C17orf98
rs10132224	0.34489928	0.015641622	8.44E-06	0.232495578	5.37E-10	-0.167638824	-6.207813079	8.24E-07	RP4-555D20.4
rs2283583	0.779399559	0.005146092	3.74E-05	0.190108033	6.18E-10	-0.149866214	-6.185657772	8.96E-07	CTD-3057O21.1

rs111419523	0.780513579	-0.004331138	6.90E-06	0.276741747	6.84E-10	-0.195712173	-6.169635416	9.39E-07	RNA5SP154
rs149065308	0.312030843	0.019666065	1.59E-06	0.348472771	9.19E-10	-0.228607193	-6.122837408	1.20E-06	CTC-537E7.3
rs11588833	0.551675394	-0.007588428	0.000416909	0.143207006	1.09E-09	-0.132385745	-6.095348091	1.29E-06	SERPINA2P

5. I would be interested to know the overlap with the other major array-based eQTL study in intestinal biopsies (PMID:29930244), as well as overlap with whole blood eQTLs (which are often used as an eQTL source for IBD).

Authors' response:

*We thank the reviewer for this suggestion. We overlapped cis-eQTL pairs from this study (FDR <0.05) with significant cis-eQTL pairs (FDR <0.05) from the CEDAR cohort study¹⁰, which provides array-based intestinal cis-eQTLs from healthy people. We compared the effect directions, replication rates are 92.86% (65 out of 70 in ileum), 92.25% (119 out of 129 in transverse colon) and 94.64% (106 out of 112 in rectum). We then overlapped our cis-eQTLs (FDR <0.05) with eQTLGen¹⁰ (FDR <0.05), which contains the blood cis-eQTLs from a large meta-analysis consisting of 31,684 population-based individuals. The replication rate is 81.44%. This suggests that we have robust intestinal cis-eQTL findings in our study and also, tissue-specific cis-eQTL effect exist. We adjusted the manuscript and added **Supplementary Figure 2A~G**.*

Results: line: 119-134

*We first compared each significant gene–single nucleotide polymorphism (SNP) pair (FDR<0.05) from our dataset with all significant cis-eQTL results from the healthy intestinal datasets of the GTEx project¹² and the ‘CEDAR’ cohort study¹³. GTEx provides three different sources of RNA-seq-based intestinal cis-eQTLs: sigmoid, transverse colon and terminal ileum. Our eQTLs overlap with 97.48% (1,085 out of 1,113) of the GTEx sigmoid cis-eQTLs, with 99.27 % (1,778 out of 1,791) of the GTEx transverse colon cis-eQTLs and 99.15% (937 out of 945) of the GTEx terminal ileum cis-eQTLs (**Supplementary Figure 2A, B, C**). The replication rates in the ‘CEDAR’ cohort study which provides three array-based intestinal cis-eQTLs datasets, are 92.86% (65 out of 70 in ileum), 92.25% (119 out of 129 in transverse colon) and 94.64% (106 out of 112 in rectum) (**Supplementary Figure 2D, E, F**). We then compared the here reported eQTLs with those found in the pediatric IBD ‘RISK’ cohort study¹⁴ (P <0.05), a targeted eQTL study on known IBD GWAS variants, we found that 83.00% of cis-eQTLs have the same direction of effect (39 out of 47, **Supplementary Table 4b**). In addition, we overlapped our cis-eQTL pairs with the findings of the eQTLGen¹⁰ meta-analysis which was*

performed on blood and found a replication rate of 81.44% (Supplementary Figure 2G), suggesting tissue-specific genetic regulatory effects to exist in our findings

Methods: line: 495-504

Comparison to other cis-eQTL data

To assess the robustness of our approach and dataset, significant cis-eQTLs (FDR <0.05) were aligned to four publicly available datasets: 1) GTEx (v7) significant cis-eQTL summary statistics¹² for Colon sigmoid (n = 124), Colon Transverse (n = 169) and Small intestine (n = 77), 2) significant intestinal eQTLs (FDR <0.05) of the 'CEDAR' study¹³ (n = 323), including ileum, transverse colon and rectum, 3) significant eQTLGen¹⁰ blood eQTLs (FDR <0.05) from 37 population-based cohorts (n = 31,684) (<https://www.eqtngen.org/>) and 4) the pediatric IBD 'RISK' cohort results¹⁴ (n = 245) with nominal P value <0.05. Proportional overlap was calculated as the proportion of significant cis-eQTLs deriving from our dataset, that replicated in these publicly available datasets with beta in the same direction.

Discussion: line: 299-301

When comparing our cis-eQTLs with those identified in larger, non-disease specific datasets, including GTEx and CEDAR studies, we found overlap of more than 92%, which supports the robustness of our findings.

Reviewer #2 (Remarks to the Author):

Thank you for inviting me to review this interesting paper by Hu et al examining cis-eQTLs in IBD using fresh frozen samples from genotyped IBD patients. The paper is interesting as it identifies potential genes that may be involved in inflammation but suffers from the very heterogenous patient and sample groups and lack of discussion about these confounders.

The paper would be improved with the following:

1. Table 1: Was any statistical analysis done to determine if any significant differences between groups

Authors' response:

*We agree with the reviewer's suggestion and performed proper tests between different groups. We used χ^2 tests to compare inflamed samples proportions between different tissue locations (colon and ileum), diagnosis (CD, UC and IBDU), sex (male and female), medication (users and non-users with mesalazines, steroids, thiopurines and anti-TNF each separately), Montreal A/L/B classifications within patients with CD, and Montreal E/S classifications within patients with UC. We used Wilcoxon tests to compare patient's age at biopsy between inflamed samples and non-inflamed samples. Only five patients used methotrexate in the whole dataset so we did not perform any analysis for methotrexate usage. We did not observe any significant differences in these comparisons ($P > 0.05$). We added the P values in **Table 1**.*

2. Table 2: Unclear why the SNPs were arranged seems to be random? ENS # - wouldn't p-value be more informative.

Authors' response:

The order of Table 2 in the manuscript was arranged by gene Ensemble IDs. We do agree with the reviewer and re-arranged Table 2, ordered by P values.

3. Samples inflamed vs non-inflamed: unclear which samples are from which patient - were these matched-pairs from the same patient? How do you define inflamed vs non-inflamed - pathologic scoring vs endoscopy? What is the definition of "lightly inflamed" - I have never heard of this term.

Authors' response:

74 samples have paired samples (inflamed and non-inflamed) but from different tissue locations. To achieve more power for cis-eQTL identification, we combined samples adjusting for repeat measurements in our analysis. The match-pairs information along with complete phenotype for each sample will be uploaded to EGA (<https://ega-archive.org/studies/EGAS00001002702>, "Multi-omics data of 1000 Inflammatory Bowel Disease patients") once the manuscript is accepted.

We removed the eight samples that were scored as lightly-inflamed and re-run all the analysis. Disease activity was registered during endoscopy based on internationally accepted scoring methods used in endoscopic evaluation of IBD¹⁵. Inflamed mucosa in all scoring methods is defined as redness, oedema and/or aphthous or deep ulceration of the mucosa, which is how we also defined inflamed mucosa. The disease specific scores are based on scoring of multiple segments. We adjusted the text accordingly:

Methods: line 400-402:

Macroscopic inflammation status was classified based on the aspect of the mucosa during colonoscopy; inflamed defined as redness and edema with or without ulceration of the mucosa¹⁵.

4. Patient - there are too many variables in this study - age, both CD and UC. Is there a possibility to use healthy controls? Not sure I understand how 80% of patients in the inflamed group are <40 years of age but the average age is 42? What is the age range of the other 20%.

Authors' response:

We are happy to clarify these points raised by the reviewer:

*The specific question we wanted to answer with this study was whether there are inflammation-specific eQTLs. Since this inflammation is only present in IBD, the comparison between inflamed and non-inflamed expression patterns can only be made within IBD samples. We do however put our findings in perspective with data from non-diseased intestinal tissues derived from the GTEx and the eQTLgen databases (**Methods: lines 496-502**).*

In Table 1 the category "A2: 17-40 years" is part of the Montreal classification for Crohn's disease, reflecting the age at diagnosis, which provides information on the disease phenotype of these patients. This does not reflect their age at time of biopsy; biopsies were generally taken many years after the first diagnosis.

5. Sample location and drug responses: how is disease location and multiple drug handled, drugs such as biologics must play a role in inflammation.

Authors' response:

We thank the reviewer for this comment. For disease location, we performed principle component analysis (PCA) and then correlated the PCs to all covariates including disease location (colon and ileum) using Spearman correlation. We found that biopsy location was most strongly correlated with the first PC ($r^2 = 0.64$, P value = $2.64e-34$, **Supplementary Figure 6A and B**). Therefore, we corrected for the 1st PC stands for the tissue location effect on the gene expression. By furthering validation based comparison gene expressions explicitly conditioning on confounders (**See minor comment 2 from reviewer 1**), we do find high consistency. For multiple drugs, we selected five medications commonly used in patients with IBD, including mesalazines, steroids, thiopurines, methotrexate and anti-TNF. However, due to very-low number of methotrexate users ($n = 5$), we removed these in our statistical analysis. We correlated the four medications with PCs and found they do have an effect on gene expressions (**Supplementary Figure 6B**). In our analysis, we corrected for the first 18 PCs explained ~77% of the gene expression variation which capture the majority of confounding effects.

6. Functional analysis: there are a number of interesting genes that are superficially discussed. Some further functional studies of the candidates with greatly increase enthusiasm for this paper.

Authors' response:

We thank the reviewer for the suggestion. We extended the results interpretation with more gene functional annotation and literature search.

Results: line: 137-145

After a heterogeneity test between these eQTL pairs and the three GTEx gut datasets, four eQTL pairs showed significance, which suggests that these intestinal cis-eQTLs indeed have a different direction of effect in the context of IBD (Q test $P < 0.05$, **Supplementary Table 4c, Supplementary Figure 3**). These four eGenes consist of: PPP2R2D, a gene involved in the cell cycle by controlling mitosis entry and exit; RBL2, a gene associated with type 2 diabetes; LIMD1, a gene involved in several cellular processes including cell-cell adhesion and cell development and ZNF593, which modulates DNA binding. Neither the eGenes nor the eSNPs have previously been reported to be associated with IBD risk.

Results: line: 151-165

At a posterior probability threshold of having one shared causal variant (PP4) of > 0.5, we discovered 558 colocalizing variants (Supplementary Table 5). For example, our IBD-based dataset showed 172 eSNPs that colocalized with IBD. The eGene that most strongly colocalized is HNF4A (PP4 =0.99), the expression of which is known to be decreased in the intestinal mucosa in patients with IBD and UC^A. Functional enrichment analysis showed that the eGenes that colocalized with IBD GWAS loci are enriched for the 'Olfactory signaling pathway' (P value = 1.4e-08) and 'G alpha (s) signaling events' (P value = 1.6e-07). Both of these pathways are forms of G protein-coupled receptor signaling, which is a basic mechanism in the immune response in IBD⁵. For colon cancer, we found four colocalizing eSNPs and for diverticulitis we found one colocalizing eSNP. 102 eSNPs colocalize with celiac disease, which are enriched for the 'ER-Phagosome pathway' (P value = 3.3e-06) and 'Nucleotide excision repair' (P value = 7.5e-06). ER stress pathways are known to play a central role in IBD inflammation⁶. These results suggest that a large part of intestinal eSNPs are likely to be causal variants in IBD and coeliac disease.

Results: line: 176-178

Amongst these eGenes are MIR214, associated with progression of UC¹⁶, C6, a complement protein encoding gene, and the gene encoding FOLR3, an anti-microbial and anti-tumor functioning protein¹⁷.

Results: line: 199-203

For example, a cis-eQTL effect between rs6860770 (A/C, $FDR_{genotype}=2.26e-16$) and gene C6, a complement protein encoding gene which plays a role in the innate and adaptive immune response¹⁸, is observed in the whole sample. However, the effect size is different in inflamed tissue compared with non-inflamed tissue ($FDR_{interaction}=0.014$, **Figure 1C**).

1. Viechtbauer, W. Conducting Meta-Analyses in R with the **metafor** Package. *J. Stat. Softw.* **36**, 1–48 (2010).
2. De Lange, K. M. *et al.* Genome-wide association study implicates immune activation of multiple integrin genes in inflammatory bowel disease. *Nat. Genet.* **49**, 256–261 (2017).
3. Giambartolomei, C. *et al.* A Bayesian framework for multiple trait colocalization from summary association statistics. *Bioinformatics* **34**, 2538–2545 (2018).
4. Ahn, S. H. *et al.* Hepatocyte nuclear factor 4 α in the intestinal epithelial cells protects against inflammatory bowel disease. *Inflamm. Bowel Dis.* **14**, 908–920 (2008).
5. Zeng, Z. *et al.* Roles of G protein-coupled receptors in inflammatory bowel disease. *World Journal of Gastroenterology* vol. 26 1242–1261 (2020).
6. Kaser, A. & Blumberg, R. S. Endoplasmic reticulum stress and intestinal inflammation. *Mucosal Immunology* vol. 3 11–16 (2010).
7. Aran, D., Hu, Z. & Butte, A. J. xCell: Digitally portraying the tissue cellular heterogeneity landscape. *Genome Biol.* **18**, (2017).
8. van der Wijst, M. G. P. *et al.* Single-cell RNA sequencing identifies celltype-specific cis-eQTLs and co-expression QTLs. *Nat. Genet.* **50**, 493–497 (2018).
9. Jenkins, C., Hewamana, S., Krige, D., Pepper, C. & Burnett, A. Aminopeptidase inhibition by the novel agent CHR-2797 (tosedostat) for the therapy of acute myeloid leukemia. *Leuk. Res.* **35**, 677–681 (2011).
10. Vösa, U. *et al.* Unraveling the polygenic architecture of complex traits using blood eQTL metaanalysis. *bioRxiv* **18**, 447367 (2018).
11. Cai, G. *et al.* CD160 inhibits activation of human CD4⁺ T cells through interaction with herpesvirus entry mediator. *Nat. Immunol.* **9**, 176–85 (2008).
12. Aguet, F. *et al.* Genetic effects on gene expression across human tissues. *Nature* **550**, 204–213 (2017).
13. Momozawa, Y. *et al.* IBD risk loci are enriched in multigenic regulatory modules encompassing putative causative genes. *Nat. Commun.* **9**, 2427 (2018).
14. Marigorta, U. M. *et al.* Transcriptional risk scores link GWAS to eQTLs and predict complications in Crohn’s disease. *Nat. Genet.* **49**, 1517–1521 (2017).
15. Marshall, J. K. *et al.* Incidence and Epidemiology of Irritable Bowel Syndrome After a Large Waterborne Outbreak of Bacterial Dysentery. *Gastroenterology* **131**, 445–450 (2006).

16. Polytarchou, C. *et al.* MicroRNA214 Is Associated With Progression of Ulcerative Colitis, and Inhibition Reduces Development of Colitis and Colitis-Associated Cancer in Mice. *Gastroenterology* **149**, 981–992.e11 (2015).
17. Holm, J. & Hansen, S. I. Characterization of soluble folate receptors (folate binding proteins) in humans. Biological roles and clinical potentials in infection and malignancy. *Biochimica et Biophysica Acta - Proteins and Proteomics* vol. 1868 (2020).
18. Ambrose, R. L., Liu, Y. C., Adams, T. E., Bean, A. G. D. & Stewart, C. R. C6orf106 is a novel inhibitor of the interferon-regulatory factor 3– dependent innate antiviral response. *J. Biol. Chem.* **293**, 10561–10573 (2018).

REVIEWERS' COMMENTS

Reviewer #1 (Remarks to the Author):

I thank the authors for the extra analyses that they have carried out, and in particular for the clarifications to the methods and the associated code, which I feel improves the manuscript. They have addressed all of my comments.

Reviewer #2 (Remarks to the Author):

The authors have responded to my comments.No further comments.